# Rapid traversal of vast chemical space using machine learning-guided docking screens

Andreas Luttens [1,2,3] ✉, Israel Cabeza de Vaca[1], Leonard Sparring[1], José Brea[4,5], Antón Leandro Martínez [4,5], Nour Aldin Kahlous [1], Dmytro S. Radchenko [6], Yurii S. Moroz [6,7,8], María Isabel Loza [4,5] ✉, Ulf Norinder [9] ✉ & Jens Carlsson [1] ✉

The accelerating growth of make-on-demand chemical libraries provides unprecedented opportunities to identify starting points for drug discovery with virtual screening. However, these multi-billion-scale libraries are challenging to screen, even for the fastest structure-based docking methods. Here we explore a strategy that combines machine learning and molecular docking to enable rapid virtual screening of databases containing billions of compounds. In our workflow, a classification algorithm is trained to identify top-scoring compounds based on molecular docking of 1 million compounds to the target protein. The conformal prediction framework is then used to make selections from the multi-billion-scale library, reducing the number of compounds to be scored by docking. The CatBoost classifier showed an optimal balance between speed and accuracy and was used to adapt the workflow for screens of ultralarge libraries. Application to a library of 3.5 billion compounds demonstrated that our protocol can reduce the computational cost of structure-based virtual screening by more than 1,000-fold. Experimental testing of predictions identified ligands of G protein-coupled receptors and demonstrated that our approach enables discovery of compounds with multi-target activity tailored for therapeutic effect.

The number of possible drug-like molecules has been estimated to be more than $10^{60}$, which exceeds the size of chemical libraries evaluated in early drug discovery by many orders of magnitude[1]. In fact, only ~13 million compounds are currently available in-stock from chemical suppliers, which clearly illustrates the limited coverage of chemical space[2]. Advances in synthetic organic chemistry have provided access to increasingly larger compound collections and make-on-demand libraries currently contain >70 billion readily available molecules[3,4].

The diverse scaffolds available in these libraries represent a major opportunity for drug discovery, but identifying the compounds relevant for a specific target in this enormous chemical space remains a major challenge.

Recently, structure-based virtual screens of ultralarge libraries have identified ligands of important therapeutic targets, demonstrating that expanding the coverage of chemical space can accelerate early drug discovery[5–7]. The most recently published docking screens

[1]Science for Life Laboratory, Department of Cell and Molecular Biology, Uppsala University, BMC, Uppsala, Sweden. [2]Infectious Disease and Microbiome Program, Broad Institute of MIT and Harvard, Cambridge, MA, USA. [3]Institute for Medical Engineering and Science and Department of Biological Engineering, Massachusetts Institute of Technology, Cambridge, MA, USA. [4]Innopharma Drug Screening and Pharmacogenomics Platform, BioFarma research group, Center for Research in Molecular Medicine and Chronic Diseases (CiMUS), Department of Pharmacology, Pharmacy and Pharmaceutical Technology, University of Santiago de Compostela, Santiago de Compostela, Spain. [5]Health Research Institute of Santiago de Compostela, Santiago de Compostela, Spain. [6]Enamine Ltd, Kyiv, Ukraine. [7]Taras Shevchenko National University of Kyiv, Kyiv, Ukraine. [8]Chemspace LLC, Kyiv, Ukraine. [9]Department of Pharmaceutical Biosciences, Uppsala University, Uppsala, Sweden. ✉e-mail: aluttens@mit.edu; mabel.loza@usc.es; ulf.norinder@uu.se; jens.carlsson@icm.uu.se

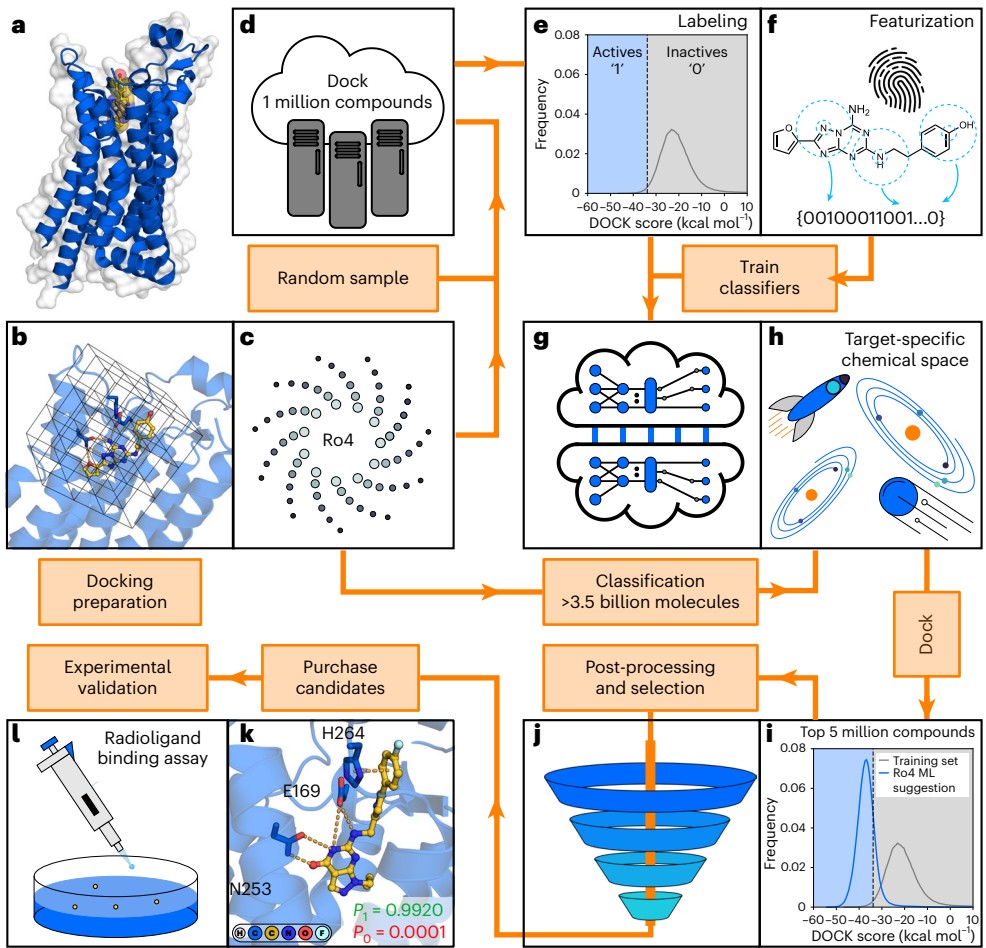

**Fig. 1 | Machine learning-accelerated virtual screening workflow. a**, Selection of a target protein for virtual screening. **b**, Model representation of protein binding site for molecular docking calculations. **c**, A subset from an ultralarge (Ro4, rule-of-four) chemical library is extracted and prepared for docking screens. **d**, Docking scores for compounds in the training set are generated. **e**, A docking score threshold splits the training set into virtual actives (1 class) and inactives (0 class). **f**, Molecules are represented by descriptors (for example, fingerprints). **g**, Machine learning models are trained to distinguish virtual actives from inactives. **h**, The trained models are used to identify a subset of predicted virtual actives in the ultralarge library. **i**, A set of compounds is selected for docking to the target. Rule-of-four molecules suggested by machine learning (Ro4 ML) have an improved docking score distribution compared to random molecules in the training set. **j**, Post-processing of docking results and selection of compounds. **k**, Compounds are selected based on visual inspection. **l**, Experimental evaluation of synthesized compounds.

have reached billions of compounds, but evaluation of these massive libraries is demanding due to the substantial computational resources required[8,9]. The make-on-demand databases will also continue to grow and probably reach trillions of compounds in the near future, which will be unfeasible to screen even with the fastest structure-based docking algorithms. Therefore, there is an urgent need for more efficient virtual screening approaches able to evaluate these vast chemical libraries[10].

Recent breakthroughs in artificial intelligence have revived interest in using quantitative structure–activity relationship (QSAR) models in drug discovery. QSAR has been widely used by the pharmaceutical industry to predict both on- and off-target activities, as well as physicochemical and pharmacokinetic properties[11]. By representing compounds using molecular descriptors, machine learning methods can rapidly evaluate large compound databases. Traditionally, QSAR models have been trained on experimental data[12], but there is an increasing interest in predicting which compounds in make-on-demand libraries are likely to receive favorable scores from computationally expensive virtual screening methods[13–15]. This combination of machine learning and molecular docking screening has the potential to enable virtual screens of multi-billion-scale compound libraries at a modest computational cost.

In this work, we developed an ultrafast workflow based on conformal prediction (CP) for screening of vast chemical libraries. The CP framework can be applied to any machine learning classifier and allows the user to control the error rate of the predictions[16,17]. Mondrian conformal predictors provide class-specific confidence levels that ensure validity for both the majority and minority class. This approach is therefore well suited for handling inherently imbalanced datasets such as virtual screening applications, which focus on identifying a small number of top-scoring compounds in a chemical library[18]. The framework has been utilized in QSAR applications to predict pharmacokinetic properties and bioactivity[19,20]. Strategies to improve the virtual screening efficiency using the CP framework have been explored, but these workflows did not achieve the efficiency required to evaluate multi-billion-scale libraries[21]. Applications of more recently developed techniques such as gradient boosting, deep neural networks and transformers to early-phase drug discovery have been successful[22–24]. Here we combined the CP framework with several state-of-the-art classification algorithms to develop a workflow for accelerated structure-based virtual screening. Our most efficient protocol identifies the top-scoring compounds in ultralarge compound libraries and reduces the number of molecules to be explicitly

docked by three orders of magnitude. We show that application of machine learning to guide docking screens of multi-billion-scale compound databases enables efficient discovery of ligands targeting G protein-coupled receptors, which is one of the most important families of drug targets[25]. In particular, our workflow can screen billions of compounds against several targets to identify ligands with activity at multiple targets relevant for the same disease.

## Results

The protocol combining CP and molecular docking to navigate ultralarge compound libraries is described in 'Machine learning-accelerated virtual screening pipeline' in Methods (Fig. 1, Supplementary Section 1 and Supplementary Fig. 1). In the development of this approach, we first conducted benchmarking docking screens against eight protein targets. The resulting datasets were used to select suitable algorithms and molecular descriptors. In the second step, the method was further optimized to enable virtual screens of multi-billion-scale libraries and applied to predict ligands of the $A_{2A}$ adenosine ($A_{2A}$R) and $D_2$ dopamine ($D_2$R) receptors.

### Benchmarking of conformal predictors

Molecular docking screens against eight therapeutically relevant proteins were carried out to initiate performance evaluation of the CP workflow. A detailed description of the protein targets and preparation of the molecular docking calculations is provided in Supplementary Table 1, Supplementary Fig. 2 and 'Preparation of proteins for docking' in Methods. Eleven million randomly sampled rule-of-four (Ro4, molecular weight <400 Da and cLog$P$ < 4) molecules from the Enamine REAL space were prepared for molecular docking and screened against each target. In total, more than 493 trillion protein–ligand complexes were predicted, resulting in a final benchmarking set of 88 million unique protein–ligand complexes and their corresponding scores. For each target, chemical structures of the compounds and their corresponding docking scores were used to create training ($10^6$ compounds) and test ($10^7$ compounds) sets for evaluating the CP framework. The energy threshold for the active (minority) class was determined based on the top-scoring 1% of each screen.

For each protein target, we assessed the performance of three different machine learning algorithms: CatBoost[26], deep neural networks[27] and Robustly Optimized Bidirectional Encoder Representations from Transformers Approach (RoBERTa)[28]. To explore diverse representations of small molecules, we trained our algorithms on three different types of features: (1) Morgan2 fingerprints, the RDKit implementation of the substructure-based ECFP4 descriptor[29], which have consistently performed among the best features in previous virtual screening benchmarks[30]; (2) recently developed continuous data-driven descriptors (CDDD)[31], which provide dense latent representations of molecules; and (3) transformer-based descriptors derived from a pretrained RoBERTa encoder, which serve as the input for fine-tuning the RoBERTa models[28]. Detailed descriptions of the hyperparameters used in the training of each classifier are provided in Supplementary Section 2 (Supplementary Table 2 and Supplementary Figs. 3–5).

Five independent classifiers were trained on 1 million labeled features, of which 80% was used for proper training and the remaining 20% for calibration. The features of the compounds in the test set (10 million compounds) were then assigned 10 normalized $P$ values (five $P_1$ and five $P_0$ values) by using each individual classification model and its corresponding calibration sets. The two resulting sets of $P_1$ and $P_0$ values were aggregated into a single pair of $P_1$ and $P_0$ values by taking the respective medians (Supplementary Fig. 1). On the basis of the aggregated $P$ values and a selected significance level ($\varepsilon$), the Mondrian CP framework was used to divide the compounds into virtual active, virtual inactive, both (meaning, either virtual active or inactive) and null (no assignment) sets (Fig. 2a). The performance on the benchmarking set was assessed using the significance level at which

the predictor resulted in the maximal number of useful (single label) predictions, $\varepsilon_{opt}$ (Fig. 2b). The metrics to assess the performance of the models (sensitivity, precision, efficiency and prediction error rate) are defined in 'Training and evaluation of machine learning classifiers' in Methods. Following the CP framework, exchangeability between the training and test sets led to strong agreement between the prediction error rate and the selected significance level[16] (Fig. 2c). To minimize the number of compounds requiring explicit docking while maximizing predictive power, we aimed to determine the optimal size of the training set, exploring a range between 25,000 and 1 million compounds. Improved sensitivity, precision, and significance values were obtained for all targets when increasing the training set size (Fig. 2d–f and Supplementary Tables 3–8).

As the performance of the models stabilized at a training size of 1 million molecules, this size was established as the standard for the training of new models. Conformal predictors composed of CatBoost classifiers trained on Morgan2 fingerprints achieved the best average precision and had comparable or slightly better significance and sensitivity values compared with other combinations. In addition, this configuration required the least computational resources, both in the training of the classifier, predictions for the test set and storage of molecular descriptors (Supplementary Table 9). Hyperparameters (class imbalance and number of aggregated models), robustness (influence of noise and scrambling of the training data), target dependency and the exchangeability criterion were also investigated. These results are provided in detail in Supplementary Figs. 6–12.

### Optimized workflow for ultralarge chemical libraries

To optimize the performance of the workflow for ultralarge databases, we conducted further analyses of datasets containing docking scores for 235 million compounds from the ZINC15 library[32], focusing on two benchmarking proteins ($A_{2A}$R and $D_2$R). For each target, a conformal predictor composed of five independent CatBoost classifiers was trained on 1 million compounds (Morgan2 representation), followed by predictions for the remainder of the library. As the docking scores were available for all the compounds in the library, efficient strategies to identify the top-scoring molecules could be established.

In the CP framework, the selected significance level determines the size of the predicted virtual active set, which is the library that will be docked to the target. The significance level was first set to achieve the maximal efficiency ($A_{2A}$R $\varepsilon_{opt}$ = 0.12 and $D_2$R $\varepsilon_{opt}$ = 0.08), and close to all compounds received a single label (>98% for both targets). CP reduced the ultralarge library from 234 million to 25 million and 19 million compounds for $A_{2A}$R and $D_2$R, respectively, with high sensitivity values (0.87 and 0.88, respectively; Fig. 3a). The workflow would hence be able to identify close to 90% of the virtual actives by docking only ~10% of the ultralarge library and the CP framework guaranteed that the percentage of incorrectly classified compounds did not exceed 12% and 8% respectively. For libraries of this size, molecular docking screens of the entire predicted virtual active set would be viable. However, further reduction of the database would be required to apply our workflow to multi-billion-scale libraries. In these cases, docking calculations for even a small percentage of the library would be computationally demanding. In theory, decreasing the significance level should lead to a reduction of the virtual active set and enrich predictions in which the conformal predictor has the highest confidence. This approach was evaluated by gradually reducing the significance level and assessing how the distribution of docking scores in the virtual active set was influenced. As anticipated, lowering the significance level did reduce the virtual active set size (Fig. 3a) and led to substantial shifts of the docking score distribution toward better energies for both protein targets (Fig. 3b). At the lowest evaluated significance level (0.01), the database was reduced to 3.0 million and 2.6 million molecules for $A_{2A}$R and $D_2$R, respectively, and the largest shifts in docking score distributions were obtained. For example, the

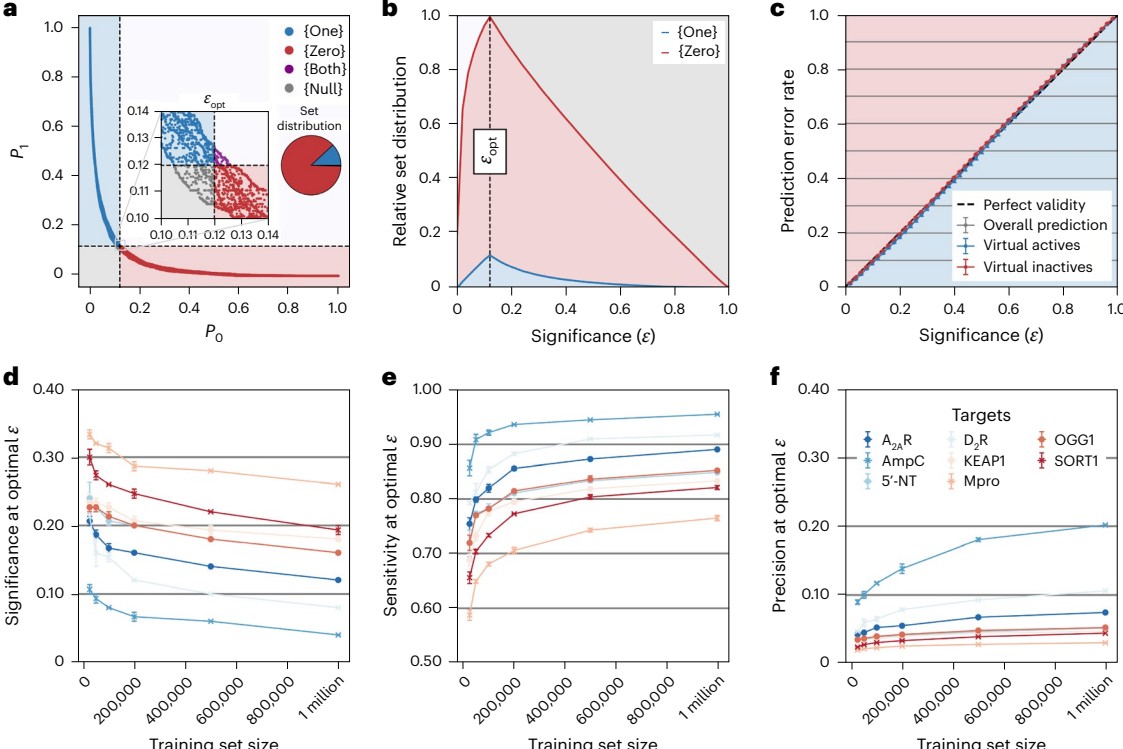

**Fig. 2 | Benchmarking of conformal predictors. a–c**, Summary of application of the Mondrian CP framework to one of the targets in the benchmarking set ($A_{2A}R$). **a**, Molecules were classified into four distinct sets based on their $P$ values and a selected significance threshold ($\varepsilon$): virtual actives (blue, 1 class), virtual inactives (red, 0 class), both (purple, 1 or 0 class) and null (gray, no class assignment). Relative fractions of each set are represented by a pie chart. **b**, The $A_{2A}R$ test set molecules were divided into four prediction sets depending on the significance level. The optimal significance ($\varepsilon_{opt}$) corresponds to the value at which the maximal number of compounds has been assigned to a single-label set (meaning, either virtual actives or inactives), referred to as maximal efficiency. **c**, The error rate (overall, virtual actives and virtual inactives) obtained for predictions of the

$A_{2A}R$ benchmarking set compounds with respect to the significance threshold (calibration plot). There was a close agreement between the significance value and the prediction error rate. **d**, The optimal significance level improved if the classification models were trained on larger datasets. **e**, At optimal efficiency, the sensitivity values improved with increasing size of the training set. **f**, At optimal efficiency, the precision values improved with increasing training set size. In **d**, **e** and **f**, three independent calculations (training and prediction) were performed for the eight targets ($A_{2A}R$, AmpC, 5'-NT, $D_2R$, KEAP1, $M_{pro}$, OGG1, SORT1; described in Supplementary Section 1) and error bars correspond to the standard error of the mean.

most populated bin in the docking score distribution for the training set was $-23.8$ kcal mol$^{-1}$ for $D_2R$, which was improved to $-47.7$ kcal mol$^{-1}$ and $-50.9$ kcal mol$^{-1}$ for significance levels of 0.08 ($D_2R \varepsilon_{opt}$) and 0.01, respectively. At the strictest significance level (0.01), 80% and 64% of the 10,000 top-scoring $A_{2A}R$ and $D_2R$ molecules (corresponding to 0.004% of the chemical library) could still be identified. These results showed that the significance level can be tuned to achieve substantial database reduction and retain most of the very top-scoring candidates for the subsequent docking step.

An alternative approach to reduce the size of the set to evaluate by molecular docking is to sort the compounds based on the difference between the $P_1$ and $P_0$ values (the quality of information, $P_1 - P_0$). This metric can be used to prioritize subsets in which the predictor has the highest confidence (Fig. 3c) and correlated with docking ranks (Supplementary Fig. 13). The enrichment of the top-scoring 10,000 molecules from the $A_{2A}R$ and $D_2R$ screens was assessed based on prioritizing the compounds using the quality of information. Remarkably, the workflow identified more than 90% of the very top-scoring molecules after only 3% ($A_{2A}R$) and 5% ($D_2R$) of the 234 million remaining compounds had been evaluated (Fig. 3c). Notably, reproducible recall values were obtained by independently generated conformal predictors, demonstrating that a random selection of training set will lead to similar selection of molecules. Data-dimensionality reduction (Uniform Manifold Approximation and Projection, UMAP) of the Morgan2 fingerprints indicated that these prioritized molecules bear

structural similarity to the actives present in the training set (Fig. 3d). This observation was also supported by analysis of Tanimoto similarity. The molecules in which the predictor had higher confidence generally showed greater structural similarity to actives from the training set (Fig. 3e). Using the quality of information to reduce the docked set of compounds hence had a similar effect to decreasing the significance level, and these two techniques can be combined in prospective screens of multi-billion-scale libraries. To assess whether the use of conformal predictors leads to a reduction in structural diversity among prioritized molecules compared with large-scale docking screens, we analyzed the 1% top-ranked molecules from both approaches for $D_2R$. Although a smaller fraction of the 1% top-ranked compounds prioritized by the conformal predictor had unique Bemis–Murcko scaffolds (13% compared with 23% from docking), a pairwise Tanimoto coefficient analysis demonstrated that the decorated versions of these scaffolds were not significantly less diverse than those identified by molecular docking alone (Supplementary Fig. 14). Collectively, these results demonstrated that top-scoring compounds could be identified using the conformal predictor. To assess its ability to find experimentally confirmed actives, known $A_{2A}R$ and $D_2R$ ligands from the ChEMBL database[33] were evaluated. Models trained only on docking data correctly classified 92% and 86% of these ligands as virtual actives. This highlights the importance of benchmarking against known actives to validate workflows before conducting prospective virtual screens (Fig. 3f).

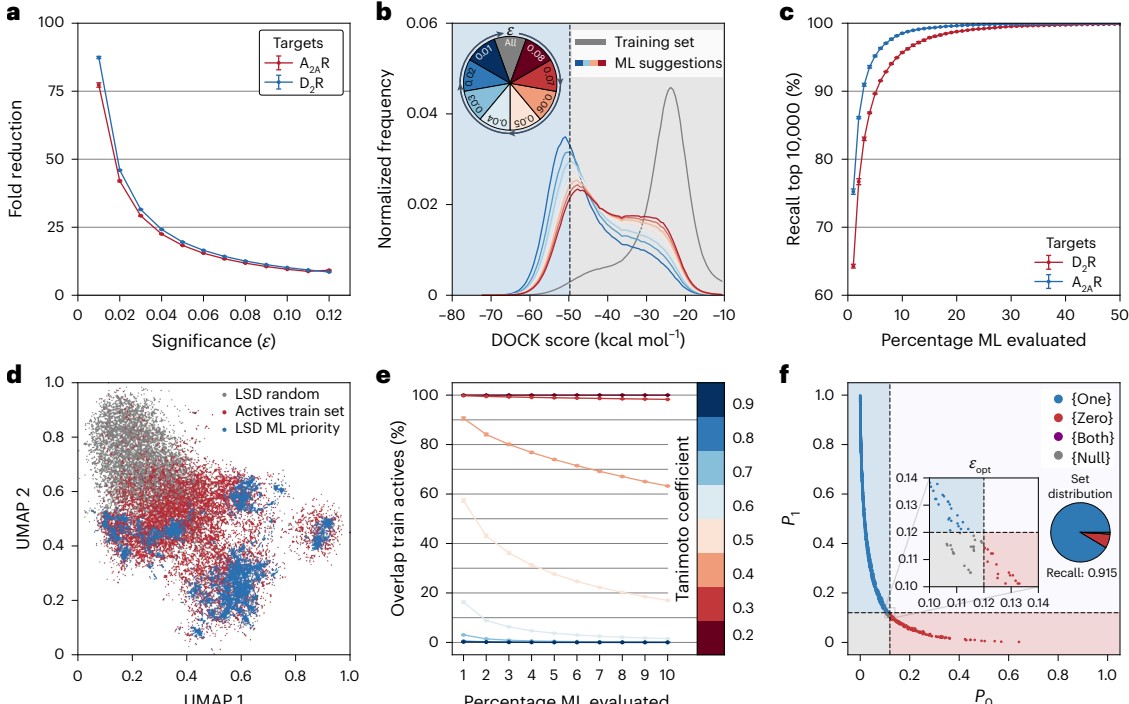

**Fig. 3 | Machine learning performance for ultralarge docking screening data.** Five independent CatBoost classifiers were trained on 1 million Morgan2 fingerprints from docking screens of 235 million molecules against $A_{2A}R$ and $D_2R$. **a**, The size of the predicted active class decreases with more stringent significance values. **b**, Normalized frequency distributions of DOCK scores from the ultralarge docking screen. The score distribution of the training set is shown in gray. In color (red to blue), score distributions of molecules machine learning (ML) predicted to be active at a given (increasingly stringent) significance threshold ($\varepsilon$). The pie chart represents different significance values. **c**, Molecules in the test set were sorted based on the quality of information ($P_1 - P_0$). The percentage recall of the 10,000 best-scoring molecules is shown as a function of the percentage evaluated compounds in the test set. **d**, Two-dimensional unsupervised UMAP projection illustrating the chemical relationship in high-dimensional feature space. Molecules prioritized by the machine learning models are more similar to training set actives than random molecules from a large-scale docking (LSD) library. **e**, Fraction of machine learning-prioritized molecules that have a Tanimoto coefficient higher than a specific threshold (0.2 to 0.9, color bar) with a molecule from the training set actives. Molecules in which the predictor is most confident are more similar to actives from the training set. **f**, $A_{2A}R$ ligands from the ChEMBL database were classified into four distinct sets based on their $P$ values and a selected significance threshold ($\varepsilon$): virtual actives (blue, 1 class), virtual inactives (red, 0 class), both (purple, 1 or 0 class) and null (gray, no class assignment). The percentage recall is represented by a pie chart. In **a**, **c** and **e**, three independent calculations (training and prediction) were performed, and error bars correspond to the standard error of the mean.

## Prospective virtual screen of a multi-billion-scale library

A primary goal in the development of the workflow was that the machine learning step must be able to reduce a multi-billion-scale database to a few million promising compounds, which was evaluated for $A_{2A}R$ and $D_2R$. Docking of the training set, training of the conformal predictor and predictions for 3.5 billion compounds for one target could be performed in approximately 2,500 core-hours. The significance level was set to 0.005, resulting in 25 million and 24 million predicted virtual actives for $A_{2A}R$ and $D_2R$, respectively. Of these, 5 million compounds per target were prioritized for docking calculations based on the quality of information (corresponding to a 700-fold reduction of the library), which required 10,344 core-hours per target. Compared with explicit docking of the 3.5 billion compounds, the workflow hence achieved a 568-fold reduction of compute cost. For both targets, the docking score distribution of the 5 million prioritized compounds was substantially shifted toward more favorable energies. For example, the most populated bin in the docking score distribution of the training set was −25.1 kcal mol⁻¹ for $D_2R$, which was shifted to −51.6 kcal mol⁻¹ for the predicted virtual actives (Fig. 4a). A large fraction of the predicted compounds (49%) had a docking score better than the energy threshold (−49.7 kcal mol⁻¹) used for labeling of the training set, corresponding to a 49-fold enrichment of virtual actives. Similar docking energy distributions were obtained when only 1 million predicted virtual actives were selected for molecular docking, demonstrating that users can control

the extent of database reduction and even achieve up to a 3,500-fold decrease in library size.

To assess whether ligands could be discovered using the workflow, we selected 31 top-ranked compounds from the $D_2R$ screen of 3.5 billion compounds and evaluated these in a radioligand binding assay at a concentration of 10 μM (Supplementary Table 10 and Supplementary Data). Of these, compounds **1** and **2** showed significant radioligand displacement and affinity values ($K_i$) values were determined for these $D_2R$ ligands ($K_i = 3.0$ μM and $K_i = 3.8$ μM, respectively; Fig. 4b, Supplementary Table 11 and Supplementary Fig. 15). To further characterize compounds **1** and **2**, we performed a functional assay quantifying $D_2R$-mediated changes in intracellular cAMP. Compounds **1** and **2** were full agonists of the $D_2R$ with potency values (EC₅₀) values of 10 μM and 14 μM, respectively (maximal effect, $E_{max} = 99\%$ and $E_{max} = 100\%$, relative to the maximal effect of dopamine; Supplementary Fig. 16). These results demonstrate that our protocol enables identification of starting points for drug discovery by docking only a small subset of compounds from a multi-billion-scale library.

## Machine learning-guided design of polypharmacology

One of the potential advantages of screening multi-billion-scale compound databases is that improved coverage of chemical space could enable discovery of ligands with complex properties, which may be difficult to find in smaller libraries containing only a few million molecules.

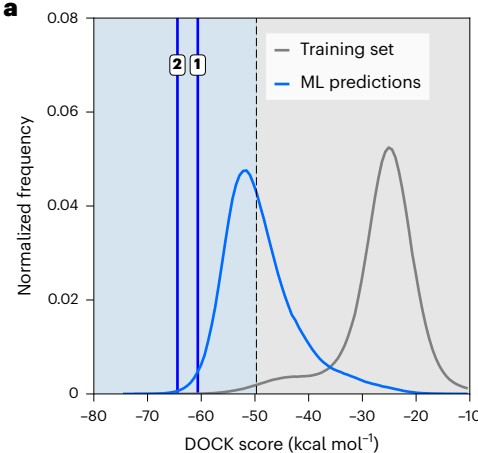

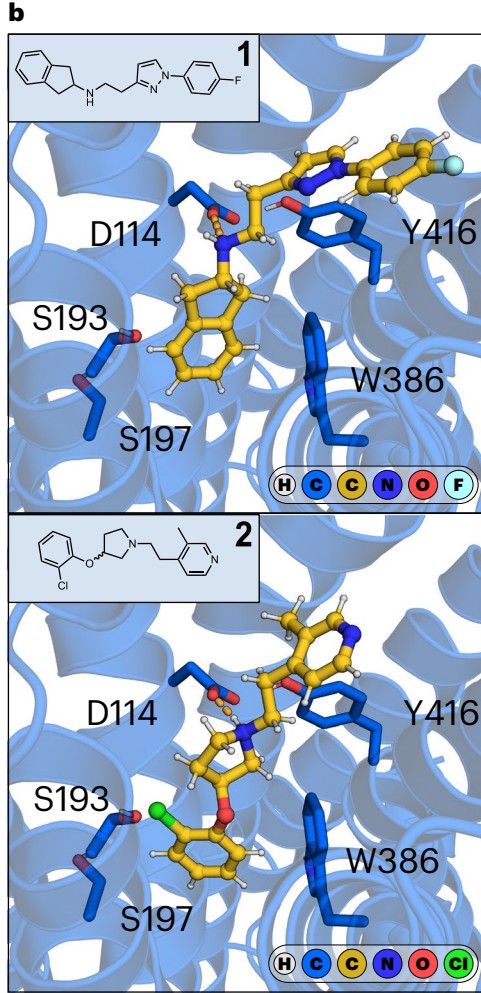

**Fig. 4 | Prospective virtual screen of a multi-billion-scale library against D₂R.** The machine learning-accelerated workflow was used to predict ligands of the D₂R in a database of 3.5 billion compounds. **a**, Normalized frequency distributions of D₂R docking scores. The docking score distribution of the training set is shown in gray. The docking score distribution of 5 million molecules selected based on the $P$ value difference ($P_1 − P_0$) is shown in blue. Vertical blue lines indicate the docking scores of the experimentally verified ligands, compounds **1** and **2**. **b**, Chemical structures and predicted binding modes of experimentally verified ligands (compounds **1** and **2**, represented as gold sticks) of D₂R (represented as a blue cartoon with side chains of key residues as sticks). Key hydrogen bonds (ionic interaction) are indicated by orange dashed lines.

One potential application is to design compounds with activity at multiple targets that are relevant for the same disease, which could lead to synergistic therapeutic effects. For example, the treatment of many central nervous system disorders requires the modulation of multiple targets (polypharmacology)[34]. Parkinson's disease is a neurological disorder that can be treated by modulating the activity of D₂R and A₂ₐR. However, the identification of dual-target ligands of A₂ₐR and D₂R is challenging because of the lack of similarity between the binding sites of the receptors[35]. To assess whether a machine learning approach could facilitate the search for dual-target ligands, A₂ₐR and D₂R were prepared for docking calculations. As the treatment of Parkinson's disease necessitates activation of D₂R through agonist binding, we generated a model of the active receptor state using homology modeling based on an agonist-bound cryogenic electron microscopy (cryo-EM) structure of the D₃ subtype (Protein Data Bank (PDB) accession code: 7CMV), which is described in detail in 'Homology modeling of active-state D₂ dopamine receptor' in Methods (Fig. 5a). To identify compounds blocking A₂ₐR signaling, we prepared an antagonist-bound crystal structure (PDB accession code: 8GNE) for molecular docking. In this structure, the salt bridge formed by the residues His264 and Glu169 has been disrupted and could potentially accommodate the ammonium cation characteristic of dopaminergic ligands. After optimizing the receptor models and docking parameters, strong enrichment of known ligands was obtained for both targets (Supplementary Fig. 17). A new training set of 1 million random molecules from a lead-like library (see 'Docking library preparation' in Methods) containing more than 3 billion compounds was then docked to both receptors. As anticipated, the resulting docking score distributions illustrate that compounds that bind to both targets would be difficult to identify in small libraries. The overlap between the top-ranked compounds (top 1%) was less than 0.02% (Fig. 5b). For both targets, conformal predictors were trained and used to predict the remaining billions of compounds in the lead-like library. To enhance the likelihood of obtaining favorable docking scores across both targets, the predicted molecules were ranked according to the sum of their quality of information ($P_{A2AR,1} − P_{A2AR,0} + P_{D2R,1} − P_{D2R,0}$: in which $P_{\varphi,1}$ and $P_{\varphi,0}$ correspond to the confidences that a specific molecule belongs to the active and inactive class, respectively, for target $\varphi$) and the 5 million top-ranked compounds were then prioritized for explicit docking. The docking score distributions of the machine learning-prioritized compounds were substantially shifted toward better energies for both targets (17- and 34-fold for A₂ₐR and D₂R, respectively; Fig. 5c), leading to an enrichment of dual virtual actives. More than 3.8% of the 5 million docked molecules had energies better than both score thresholds (top 1%), corresponding to a 191-fold enrichment of dual virtual actives compared with docking of a random library. The molecules were subsequently sorted according to the sum of their individual docking ranks (rank_A2AR + rank_D2R), and the top-ranked molecules were then visually inspected for their complementarity with the respective binding sites. Encouragingly, the molecules formed hydrogen bonds with residues known to be important for ligand binding to the A₂ₐR (Asn253[6.55]) and D₂R (Asp114[3.32]) (Ballesteros-Weinstein residue numbering scheme[36] denoted as superscripts). A set of 45 compounds was prioritized for make-on-demand synthesis and successfully obtained within 4–5 weeks. The compounds were first tested in an A₂ₐR radioligand binding assay at a concentration of 20 μM (Supplementary Table 12 and Supplementary Data). Of these, the binding affinities of four compounds (**4**–**6**) that showed significant radioligand displacement were determined, leading to $K_i$ values ranging from 1.3 μM to 20 μM (Supplementary Table 13). Compounds **4**–**6** were subsequently tested at D₂R at a concentration of 20 μM, which showed that compound **5** also binds to this target ($K_{i,D2} = 14$ μM, $K_{i,A2A} ≈ 20$ μM; Fig. 5d, Supplementary Table 14 and Supplementary Fig. 18). The dual-target compound **5** was predicted to form hydrogen bonds with orthosteric binding site residues Asn253[6.55] and Asp114[3.32] for A₂ₐR and D₂R, respectively (Fig. 5e,f). These observations indicate that our virtual screening workflow can

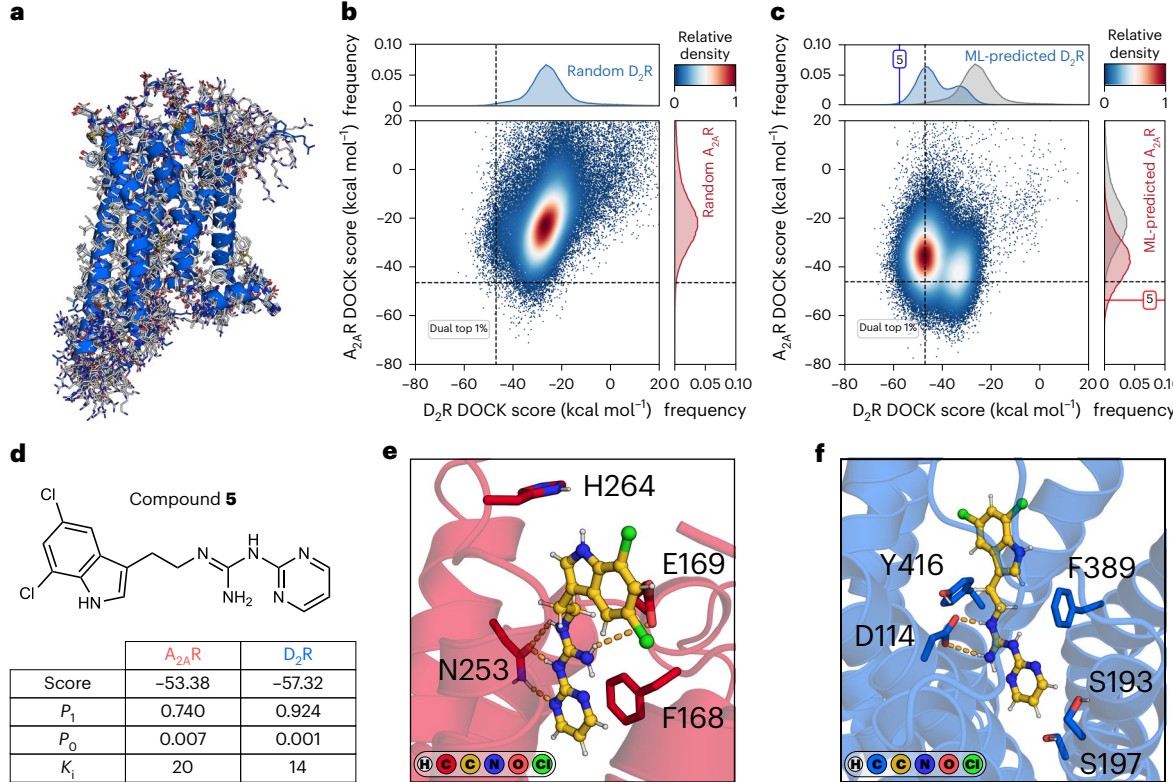

**Fig. 5 | Identification of a dual-target ligand by screening of multi-billion-scale libraries. a**, Homology models (gray lines) of the active $D_2R$ were constructed using a cryo-EM structure of the $D_3$ dopamine receptor complexed with the $G_i$ protein (PDB accession code: 7CMV) as a template. The final model used for prospective screening is depicted in blue. **b**, Two-dimensional density-normalized scatterplot of docking scores of 1 million random molecules present in the training set. Normalized frequency distribution of molecules docked against $D_2R$ (blue) and $A_{2A}R$ (red). Dashed lines represent the score threshold (top 1%) that divides the datasets into virtual actives and inactives. **c**, Two-dimensional density-normalized scatterplot of docking scores of 5 million molecules prioritized by machine learning models. For both $A_{2A}R$ and $D_2R$, 5

independent CatBoost classifiers were trained on 1 million random molecules from a docking screen. Normalized frequency distribution of molecules docked against $D_2R$ (blue) and $A_{2A}R$ (red). The score distributions of the training set compounds are shown in gray. The docking scores of compound **5** are shown as a vertical blue line for $D_2R$ and a horizontal red line for $A_{2A}R$. **d**, The chemical structure of compound **5**, a $A_{2A}R$–$D_2R$ dual-target ligand, is shown. For both targets, the docking scores (in kcal mol$^{-1}$), $P$ values (1, confidence in being a virtual active; 0, confidence in being a virtual inactive) and $K_i$ values (µM, determined from 3 independent experiments) are shown. **e**, Predicted binding mode of compound **5** in the $A_{2A}R$ orthosteric site. **f**, Predicted binding mode of compound **5** in the $D_2R$ orthosteric site.

be applied to identify starting points for the development of drugs with multi-target profiles tailored for treatment of complex diseases.

## Discussion

The rapid expansion of commercial chemical libraries has sparked the development of diverse structure-based virtual screening methods aiming to reduce the computational cost of exploring chemical space. Several of these approaches incorporated machine learning techniques to efficiently evaluate libraries ranging from millions to billions of compounds[13–15,37]. Compared with previously developed methods tackling ultralarge libraries, our approach is based on CP, a robust framework that enables control over the error rate of the predictions[16]. We extend the application of conformal predictors to multi-billion-scale libraries by leveraging class-specific confidence levels to identify top-scoring molecules. Our approach achieves equivalent or better recall values and database reductions as other workflows, without the need for resource-intensive active learning. Our comparison of substructure-based fingerprints with more recently developed data-driven descriptors demonstrates that traditional fingerprints suffice for this type of application, in agreement with a recently published study[37]. Integrating these key advantages led to a workflow capable of traversing vast chemical libraries but requiring only modest computational cost associated with the training and prediction steps. Notably, our results demonstrate that the effectiveness of machine learning as

a proxy for molecular docking is target dependent, underscoring the need for large and diverse benchmarking sets to achieve generalizable methods. To catalyze further development of methods in this field, we share both our virtual screening workflow, which is compatible with any docking software, and extensive benchmarking sets for eight diverse protein targets.

Other methods to explore large chemical libraries via molecular docking use a hierarchical approach[38,39], which is fundamentally different from the machine learning techniques. For example, the V-SYNTHES method is based on the same principles as fragment-based ligand discovery. First, a small set of fragment-sized compounds called synthons, which represent substructures of larger compounds in the multi-billion-scale library, is docked to the binding site. In a second step, larger molecules that embed top-scoring synthons are docked to identify compounds with improved docking scores. Currently, hierarchical approaches do not require the use of machine learning because the number of synthons and their corresponding elaborations are small enough to allow for explicit molecular docking. Furthermore, as the synthon library is not exchangeable with the database of fully enumerated molecules, a conformal predictor trained on synthon docking results is unlikely to accurately predict top-scoring compounds. As the libraries continue to grow, a comparison of the effectiveness of the hierarchical and machine learning approaches could reveal how these complementary techniques could be combined to further accelerate virtual screens.

A unique aspect of our work is the combination of a powerful machine-learning method with experimental evaluation of predictions, which reveals the potential and limitations of this approach. Our first screen showed that agonists of $D_2R$, an important drug target for neuropsychiatric and neurodegenerative diseases, could be identified by the workflow. Interestingly, the hit rate from the docking screen was comparable to previous screens using smaller libraries[40]. Although screens of large libraries has yielded exceptionally potent compounds, these results suggest that further progress may be partially limited by the accuracy of molecular docking. As recent studies have shown, flaws in docking scoring functions may lead to an accumulation of false positives among the top-scoring compounds in large libraries[10,41]. By reducing the number of compounds requiring explicit docking, our approach enables resources to be reallocated toward accurate re-scoring of top-ranked compounds using more advanced methods. Despite these challenges, our second prospective screen for multi-target ligands also illustrates the tremendous opportunities provided by access to larger libraries, which will soon reach the trillion scale. Encouragingly, prospective screens against $A_{2A}R$ and $D_2R$ identified a dual-target ligand, which represents a promising starting point for the development of drugs for the treatment of Parkinson's disease[34]. Hence, expanding the sampling across broader regions of chemical space can enable the discovery of ligands with complex properties that may not be found in smaller libraries. A further extension of our virtual screening approach could involve the multi-objective design of ligands with specific selectivity, physiochemical and pharmacokinetic properties by integrating conformal predictors trained for different tasks. Collectively, our results demonstrate how docking screens guided by conformal predictors can accelerate the development of small-molecule therapeutics.

## Methods

### Docking library preparation
The Enamine REAL Database (November 2019 version, 12.3 billion compounds) was reduced to a rule-of-four chemical subspace by excluding compounds with a molecular weight over 400 Da and cLog*P* over 4 using RDKit[42]. The rule-of-four subspace contained 3,541,746,925 compounds. A random sample containing 15 million compounds (0.4%) from this library was obtained after shuffling the Simplified Molecular Input Line Entry System (SMILES) with Terashuf[43]. Molecules were prepared for docking using DOCK3.7 standard protocols[44]. CXCalc (ChemAxon's Marvin package Marvin 18.10.0) was used to calculate predominant protomers at relevant pH levels (6.9, 7.4 and 7.9). Conformational ensembles were generated with OMEGA (OpenEye, version 2020.2) and were capped at 400 conformations and an inter-conformer root mean squared deviation (r.m.s.d.) diversity threshold of 0.25 Å. In-house preparation of these rule-of-four molecules approximately took 18 CPU-seconds per compound. Preparation of 1 million molecules present in our training set for docking was completed in approximately 5,000 CPU-hours. In the prospective screens for dual-target compounds, the Enamine REAL Database (November 2022 version, 33.5 billion compounds) was reduced to a subspace of 3,137,276,984 lead-like molecules (ZINC Database definition: $20 \leq$ heavy atom count $\leq 25$ and $-5 \leq$ cLog*P* $\leq 3.5$) using a similar procedure to that described above. The conformational ensembles of lead-like molecules were capped at 200 conformations and an inter-conformer r.m.s.d. diversity threshold of 0.5 Å. A random sample of 1 million rule-of-four WuXi GalaXi molecules (March 2024 version) was prepared using the same protocol as described above. The WuXi GalaXi rule-of-four space contained 1,371,598,090 compounds.

### Molecular descriptors
Canonical SMILES were used to generate three different molecular descriptors as input data for the machine learning classifiers. Morgan2 descriptors were generated using RDKit[29,42]. Continuous data-driven descriptors were generated using the CDDD Python library[31]. The RoBERTa model generated descriptors directly from the SMILES. We used a pretrained RoBERTa model[45] to generate the internal encoded representation of each molecule using the simpletransformers Python library[46].

### Preparation of proteins for docking
Experimental structures of the selected protein targets were extracted from the PDB. Details regarding preparation of crystal structures for molecular docking are provided in Supplementary Table 1. Unless stated otherwise, water molecules and other solutes were removed from the experimental structures. The N- and C-termini were capped with acetyl and methyl groups, respectively, using PyMOL[47]. The atoms of the bound ligands were used to generate matching spheres in the binding site. DOCK3.7 uses a flexible ligand algorithm that superimposes rigid segments of a molecule's pre-calculated conformational ensemble on top of the matching spheres[44]. Histidine protonation states were assigned manually after visual inspection of the hydrogen-bonding network. The remainder of the protein structure was protonated by REDUCE[48] and assigned AMBER[49] united atom charges. The dipole moments of key residues involved in recognition of the bound ligands were increased to favor interactions with these (Supplementary Fig. 2). This technique is common practice for users of DOCK3.7 to improve docking performance and has been used in previous virtual screens[50]. The atoms of the co-crystallized ligands were used to create two sets of sphere layers on the protein surface (referred to as thin spheres). One set of thin spheres described the low dielectric region of the binding site. A second set of thin spheres was used to calibrate ligand desolvation penalties. Scoring grids were pre-calculated using QNIFFT[51] for Poisson–Boltzmann electrostatic potentials, SOLVMAP[52] for ligand desolvation, and CHEMGRID[53] for AMBER van der Waals potentials. For each protein target, known ligands were retrieved from the ChEMBL database or previous studies, followed by generation of property-matched decoys according to the procedure described in ref. 54. Actives and decoys control sets were docked to the protein structures to evaluate the influence of different docking grid parameters (the radii of electrostatic and desolvation thin spheres). Finally, ligand enrichment and predicted binding poses were used to select the optimal grid parameters[50].

### Homology modeling of active-state $D_2$ dopamine receptor
Alignment of the human $D_2$ and $D_3$ dopamine receptor sequences was performed using the GPCRdb (https://gpcrdb.org/)[55]. One hundred homology models of the activated $D_2R$ bound to agonist ligand (PD128907) were constructed using MODELLER[56] version 10.2 based on a cryo-EM structure of the $D_3$ dopamine receptor complexed with the $G_i$ protein[57] (PDB accession code: 7CMV) as a template. Residues 5 to 11 were restrained to form an alpha helix and the residue pairs Cys77–Cys152 and Cys235–Cys237 were set to form disulfide bridges. Molecular docking grids of homology models were constructed using the same protocols as for the antagonist-bound $D_2$ dopamine receptor crystal structure (PDB accession code 6CM4; Supplementary Table 1). The resulting docking grids were then evaluated for their ability to reproduce the modeled binding mode of PD128907 in the corresponding homology models and enrich 25 known $D_2R$ agonists among a set of in-house-generated decoys[54]. A final receptor model was selected based on agonist enrichment calculations and the presence of the conserved salt bridge interaction with Asp114[3.32] in docking screens.

### Molecular docking calculations
The orientational sampling parameter was set to 5,000 matches for both the rule-of-four benchmarking set and molecules selected based on machine learning predictions. Molecules in the ultralarge docking screens (235 million lead-like molecules from the ZINC15 database[32]) were docked using 1,000 matches. During the generation of the benchmarking dataset, for each docked compound, 18,652 orientations were

calculated on average, and for each orientation, an average of 1,654 conformations were sampled. The best-scoring pose of each ligand was optimized using a simplex rigid-body minimizer.

### Compound selection from docking screens

To bias the multi-billion virtual screen targeting the $D_2R$ toward identification of novel chemotypes, we selected compounds that were dissimilar to known dopamine receptor ligands (>11,000 ChEMBL compounds). The molecular diversity was increased by clustering the 100,000 top-scoring docked compounds (0.003% of the entire library) by topological similarity. The best-scoring molecule from each cluster was visually inspected for its complementarity with the binding site and a set of 31 compounds were selected from the 1,000 top-ranking clusters (Supplementary Table 10 and Supplementary Data). The make-on-demand compounds were available in in the Enamine REAL Database and were successfully synthesized in 4–5 weeks.

### Machine learning-accelerated virtual screening pipeline

Our workflow for combining machine learning and molecular docking (Fig. 1) consists of the following consecutive steps, which are described in detail in this section and Supplementary Fig. 1.

**Step 1 Preparation and docking of the training set.** A set of randomly selected molecules from an ultralarge chemical library is docked to the target protein structure (Fig. 1a–d). We recommend a training set of 1 million molecules in virtual screens of multi-billion-scale libraries.

**Step 2 Generation and labeling of the training set.** A docking score threshold (Fig. 1e) is selected to label each compound in the training set as either virtual active (better score than the selected threshold) or inactive (equal or worse score than the selected threshold). As our CP approach is based on aggregating predictions made by several classifiers, multiple independent training sets are generated. Our recommendation is to label the top-scoring 1% of the training set as virtual active and generate 5 independent training sets.

**Step 3 Molecule featurization and training of the classifier.** Molecular descriptors of each molecule in the training set are generated as input for the classifier. Each of the training sets is used to train an independent classification model to distinguish virtual actives from inactives (Fig. 1f,g).

**Step 4 Conformal prediction for the ultralarge library.** The trained classification models are used to evaluate compounds from the ultralarge chemical library (Fig. 1h). Mondrian conformal predictors provide class-specific confidence levels, which allow compounds to be categorized into one of the following four sets based on a selected significance level ($\varepsilon$): virtual active, virtual inactive, both (virtual active or inactive) and null (no class assignment). The significance level can be tuned to control the size of the virtual active set, which is predicted to contain compounds with a docking score better than the selected threshold.

**Step 5 Post-processing and compound selection.** The database reduction level achieved by the workflow is target dependent, and additional post-processing steps can be applied to identify the most promising compounds. The compounds assigned to the virtual active set are ranked by sorting them based on the quality of information (meaning, prioritizing the predictions in which the classification model has the highest confidence) and a subset of these are docked to the target. Top-scoring molecules are clustered by chemical similarity and representative compounds are visually inspected (Fig. 1i,j), followed by synthesis and experimental evaluation of selected compounds (Fig. 1k–l). We recommend docking a set of 1 million to 5 million molecules selected based on the quality of information.

### Training and evaluation of machine learning classifiers

CP is a QSAR method in which an ensemble of models is used to classify molecules present in an objective set (Supplementary Fig. 1). The docking scores of the datasets were used to label molecules as virtual actives (top 1%) and virtual inactives (bottom 99%), unless stated otherwise. The scikit-learn 0.24.2 package[58] was used to perform a stratified split of the datasets into proper training sets (80% of training set), calibration sets (20% of training set) and test sets. The ratio between virtual actives and inactives was maintained in all sets. This procedure was repeated using different random seeds to obtain independent sets. The CatBoost 0.26 Python package was used for building and training the corresponding classifiers. The PyTorch 1.7.1 package[59] combined with the RangerLars optimizer[60] was used for training the deep neural networks. The RoBERTa classifier was implemented from the simpletransformers 0.61.6 package[46]. The Skorch 0.10.0 package[61] was used to connect the scikit-learn and PyTorch frameworks. A detailed description and analyses of the hyperparameters used in each classifier are provided in Supplementary Table 2 and Supplementary Figs. 3–5. The compounds in the test set were assigned normalized $P$ values (confidence that the sample belongs to 1 class, $P_1$; and 0 class, $P_0$) by each individual classification model and its corresponding calibration set. The resulting sets of $P_1$ and $P_0$ values were aggregated into a single pair of $P$ values by taking the respective medians of predictions made by individual models (Supplementary Fig. 1). On the basis of the aggregated $P$ values and the selected significance level ($\varepsilon$), the Mondrian CP framework was used to divide the compounds into virtual active, virtual inactive, both (meaning either virtual active or inactive) and null (no class assignment) sets. Several metrics were used to assess the performance of the conformal predictors. The sensitivity was defined as:

$$\text{Sensitivity} = \frac{\text{TP}}{\text{AP}} \qquad (1)$$

where TP (true positives) were true active molecules correctly classified by the CP framework (that is, in the predicted virtual active and both sets). AP (all positives) were all molecules with a score better than the threshold used to define the virtual actives. The precision was defined as:

$$\text{Precision} = \frac{\text{TP}}{\text{TP} + \text{FP}} \qquad (2)$$

where FP (false positives) were true inactive molecules incorrectly classified by the CP framework (that is, in the predicted virtual active and null sets). The efficiency was defined as:

$$\text{Efficiency} = \frac{\{1\} + \{0\}}{\text{AP} + \text{AN}} \qquad (3)$$

where {1} are the predicted virtual actives and {0} the predicted virtual inactives. AN (all negatives) were all molecules with a score worse than or equal to the threshold used to define the virtual actives. The overall error rate was defined as:

$$\text{Overall error rate} = \frac{\text{FP} + \text{FN}}{\text{AP} + \text{AN}} \qquad (4)$$

where FN (false negatives) are true virtual active molecules incorrectly classified by the CP framework (that is, in the predicted virtual inactives and null sets). The error rate for the virtual actives was defined as:

$$\text{Actives error rate} = \frac{\text{FN}}{\text{AP}} \qquad (5)$$

The error rate for the virtual inactives was defined as:

$$\text{Inactives error rate} = \frac{\text{FP}}{\text{AN}} \qquad (6)$$

## Computational costs and hardware specifications

To train a conformal predictor on 1 million Morgan2 fingerprints, approximately 4 GB of random access memory was required. Training and predicting were performed using 2x Intel Xeon Gold 6130 CPUs @ 2.10 GHz. The RoBERTa models were trained on 12 cores of a single Nvidia Tesla T4 graphics processing unit with 1,844 GiB memory. The times (in seconds) required to train a conformal predictor on 1 million molecules with different architectures and descriptors or predict 1 million molecules are reported in Supplementary Table 9.

## Binding assays

Screening compounds (Supplementary Tables 10 and 12) were purchased from Enamine (compound purity >90%, which was confirmed by liquid chromatography−mass spectrometry and $^1$H NMR spectroscopy for identified ligands; Supplementary Figs. 19−30). Human $D_2R$ competition binding experiments were carried out in polypropylene 96-well plates. Each well contained 20 μg of membranes from a CHO-$D_2R$ #S20 cell line (protein concentration of 4,322 μg ml$^{-1}$), 1.5 nM [$^3$H]-spiperone (54.3 Ci mmol$^{-1}$, 1 mCi ml$^{-1}$, PerkinElmer NET1187250UC) and the compound studied. Non-specific binding was determined in the presence of 10 μM sulpiride (Sigma Aldrich S8010). The reaction mixture (250 μl per well) was incubated at 25 °C for 120 min, after which 200 μl was transferred to a GF/C 96-well plate (Millipore) pretreated with 0.5% PEI and treated with binding buffer (50 mM Tris-HCl, 1 mM EDTA, 5 mM MgCl$_2$, 5 mM KCl, 120 mM NaCl, pH 7.4). The reaction mixture was filtered and washed 4 times with 250 μl wash buffer (50 mM Tris-HCl, 0.9% NaCl, pH 7.4) before the addition of 30 μl Universol. The final measurement was performed in a microplate beta scintillation counter (Microbeta Trilux, PerkinElmer). Human $A_{2A}R$ competition binding experiments were carried out in a multiscreen GF/C 96-well plate (Millipore) pretreated with binding buffer (Tris-HCl 50 mM, EDTA 1 mM, MgCl$_2$ 10 mM, 2 U ml$^{-1}$ adenosine deaminase, pH 7.4). Each well was incubated with 5 μg of membranes from the HeLa-$A_{2A}$ cell line and prepared in our laboratory (lot A003/14-04-2019, protein concentration 2,058 μg ml$^{-1}$), 1 nM [$^3$H]-ZM241385 (50 Ci mmol$^{-1}$, 1 mCi ml$^{-1}$, ARC-ITISA 0884), and the compounds studied and standards. Non-specific binding was determined in the presence of NECA 50 μM (Sigma E2387). The reaction mixture (total volume of 200 μl per well) was incubated at 25 °C for 30 min, then filtered and washed 4 times with 250 μl wash buffer (Tris-HCl 50 mM, EDTA 1 mM, MgCl$_2$ 10 mM, pH 7.4), and measured in a microplate beta scintillation counter (Microbeta Trilux, PerkinElmer). Unless stated otherwise, three independent experiments were carried out to calculate $K_i$ values.

## Functional assays

Human $D_2R$ activity was measured by determining the amount of cAMP produced. Human $D_2R$ functional experiments were carried out in a CHO-$D_2$ #S20 cell line. Five-thousand cells were seeded in 30 μl of Optimem (Invitrogen 11058) + 500 μM IBMX (Sigma 17018) on a 96-well black-and-white isoplate (PerkinElmer 6005030). Test compounds and dopamine were added in their corresponding wells and incubated for 10 min at 37 °C with gentle stirring (150 r.p.m.). Then, 10 μM forskolin (Sigma 17018) was added and incubated for 5 min at 37 °C with gentle stirring (150 r.p.m.). Reagents from the kit (#CISBIO 62AM4PEC) were added and incubated for 1 h at room temperature with gentle stirring (90 r.p.m.) and protected from light. HTRF (excitation wavelength: 320 nm; emission wavelengths: 620−665 nm) from each well was measured using a Tecan Infinite M1000 Pro. Two independent experiments were carried out to calculate $EC_{50}$ and $E_{max}$ values.

## Statistics and reproducibility

Compounds for which no molecular descriptors could be generated were excluded from the analyses. No statistical method was used to predetermine the sample size. Training and test sets were generated by taking random samples with the determined size of the virtual libraries.

Proper training and calibration sets were constructed by performing a stratified split on the parent training sets, maintaining the ratio of samples belong to the minority and majority classes. Unless stated otherwise, three independent calculations (training and predictions) were performed to derive statistics on model performance. Unless stated otherwise, three independent experiments were performed to derive statistics on the activities of the designed compounds.

## Reporting summary

Further information on research design is available in the Nature Portfolio Reporting Summary linked to this article.

## Data availability

The ZINC15 and Enamine REAL databases are available at https://zinc15.docking.org and https://enamine.net/compound-collections/real-compounds/real-database, respectively. The PDB accession codes for the molecular docking calculations are 4EIY, 6DPT, 6XUE, 6CM4, 5FNU, 6W63, 6G3Y, 6X48, 8GNE and 7CMV. Associated source data are provided for Figs. 2−5. All compounds tested are listed in the Supplementary Tables 10 and 12, and Supplementary Data. Chemical identities, purities (liquid chromatography−mass spectrometry), yields and spectroscopic analysis ($^1$H NMR) for active compounds are provided in the Supplementary figures. Large-scale docking datasets are deposited on Zenodo at https://doi.org/10.5281/zenodo.7953917 (ref. 62).

## Code availability

The conformal predictor source code is freely available and can be found on the GitHub at https://github.com/carlssonlab/conformal-predictor. The original code is deposited on Zenodo at https://doi.org/10.5281/zenodo.14709041 (ref. 63).

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

## Acknowledgements

A.L. was supported by a postdoctoral scholarship from the Knut and Alice Wallenberg Foundation (KAW2022.0347). J.C. received funding from the European Research Council (ERC) under the European Union's Horizon 2020 research and innovation program (grant agreement 715052), the Swedish Cancer Society, the Swedish Research Council and the Olle Engkvist Foundation. This research was partially supported by the project AI4Research at Uppsala University. I.C.d.V. was funded by a postdoctoral fellowship provided by the Sven och Lilly Lawski foundation. The computations were enabled using resources provided by the National Academic Infrastructure for Supercomputing in Sweden (NAISS) (partially funded by the Swedish Research Council through grant agreement number 2022-06725) and the supercomputing resource Berzelius provided by the National Supercomputer Centre at Linköping University and the Knut and Alice Wallenberg Foundation. J.B., A.L.M. and M.I.L. were funded by Agencia Estatal de Investigación (PID2020-119428RB-I00), Xunta de Galicia (ED431C 2022/20) and European Regional Development Fund (ERDF). A.L., I.C.d.V. and J.C. thank OpenEye Scientific Software for the use of OEToolkits at no cost. We thank J. Zhang for providing the initial deep neural network code.

## Author contributions

A.L., U.N. and J.C. designed the study. A.L. performed the molecular docking, homology modeling and machine learning calculations and selected compounds under the supervision of J.C. and U.N. A.L., I.C.d.V., L.S. and U.N. developed the protocol. A.L. and I.C.d.V. wrote the final version of the code. N.A.K. performed sequence alignments for homology modeling, constructed the $D_2R$ homology models and provided the $D_2R$ agonist set. The binding and functional assays were performed by the USEF screening platform under the supervision of J.B., A.L.M. and M.I.L. D.S.R. and Y.S.M. provided the Enamine REAL Database and analytical data for the synthesized compounds. U.N. provided support with critical evaluation of the machine learning protocol. A.L., I.C.d.V. and J.C. wrote the paper with contributions from the other authors.

## Funding

## Competing interests

J.C. is a founder of DareMe Drug Discovery Consulting. D.S.R. and Y.S.M. are employed by Enamine Ltd. Y.S.M. serves as a scientific advisor to Chemspace LLC. The other authors declare no competing interests.

## Additional information

**Correspondence and requests for materials** should be addressed to Andreas Luttens, María Isabel Loza, Ulf Norinder or Jens Carlsson.

# Reporting Summary

## Statistics

For all statistical analyses, confirm that the following items are present in the figure legend, table legend, main text, or Methods section.

| n/a | Confirmed | |
|---|---|---|
| ☐ | ☒ | The exact sample size ($n$) for each experimental group/condition, given as a discrete number and unit of measurement |
| ☐ | ☒ | A statement on whether measurements were taken from distinct samples or whether the same sample was measured repeatedly |
| ☐ | ☒ | The statistical test(s) used AND whether they are one- or two-sided *Only common tests should be described solely by name; describe more complex techniques in the Methods section.* |
| ☒ | ☐ | A description of all covariates tested |
| ☒ | ☐ | A description of any assumptions or corrections, such as tests of normality and adjustment for multiple comparisons |
| ☐ | ☒ | A full description of the statistical parameters including central tendency (e.g. means) or other basic estimates (e.g. regression coefficient) AND variation (e.g. standard deviation) or associated estimates of uncertainty (e.g. confidence intervals) |
| ☐ | ☒ | For null hypothesis testing, the test statistic (e.g. $F$, $t$, $r$) with confidence intervals, effect sizes, degrees of freedom and $P$ value noted *Give P values as exact values whenever suitable.* |
| ☒ | ☐ | For Bayesian analysis, information on the choice of priors and Markov chain Monte Carlo settings |
| ☒ | ☐ | For hierarchical and complex designs, identification of the appropriate level for tests and full reporting of outcomes |
| ☒ | ☐ | Estimates of effect sizes (e.g. Cohen's $d$, Pearson's $r$), indicating how they were calculated |

*Our web collection on statistics for biologists contains articles on many of the points above.*

## Software and code

Policy information about availability of computer code

| Data collection | Molecular modeling was performed with DOCK3.7.1, MODELLER v10.2. CXCalc (ChemAxon's Marvin package Marvin 18.10.0) was used to calculate predominant protomers at relevant pH levels (6.9, 7.4, 7.9). Conformational ensembles were generated with OMEGA (OpenEye, version 2020.2). Machine learning was performed using Python's CatBoost 0.26 package, PyTorch 1.7.1 package and the RangerLars optimizer. The RoBERTa classifier was implemented from the simpletransformers 0.61.6 package. Skorch 0.10.0 package. ZINC15 and Enamine REAL databases are available at: https://zinc15.docking.org and https://enamine.net/compound-collections/real-compounds/real-database. Software used in data generation and analysis can be found at https://github.com/carlssonlab/conformalpredictor. |
|---|---|
| Data analysis | Python libraries, RDKit 2019_Q3 and OpenEye Toolkits 2020.0.4. |

For manuscripts utilizing custom algorithms or software that are central to the research but not yet described in published literature, software must be made available to editors and reviewers. We strongly encourage code deposition in a community repository (e.g. GitHub). See the Nature Portfolio guidelines for submitting code & software for further information.

## Data

Policy information about availability of data

All manuscripts must include a data availability statement. This statement should provide the following information, where applicable:
- Accession codes, unique identifiers, or web links for publicly available datasets
- A description of any restrictions on data availability
- For clinical datasets or third party data, please ensure that the statement adheres to our policy

> Data are made freely available on Zenodo (10.5281/zenodo.7903160)

## Research involving human participants, their data, or biological material

Policy information about studies with human participants or human data. See also policy information about sex, gender (identity/presentation), and sexual orientation and race, ethnicity and racism.

| | |
|---|---|
| Reporting on sex and gender | N/A |
| Reporting on race, ethnicity, or other socially relevant groupings | N/A |
| Population characteristics | N/A |
| Recruitment | N/A |
| Ethics oversight | N/A |

Note that full information on the approval of the study protocol must also be provided in the manuscript.

# Field-specific reporting

Please select the one below that is the best fit for your research. If you are not sure, read the appropriate sections before making your selection.

☒ Life sciences          ☐ Behavioural & social sciences          ☐ Ecological, evolutionary & environmental sciences

For a reference copy of the document with all sections, see nature.com/documents/nr-reporting-summary-flat.pdf

# Life sciences study design

All studies must disclose on these points even when the disclosure is negative.

| | |
|---|---|
| Sample size | As part of the hyperparameter exploration, the number of samples in the training sets varied from 25,000 to 1,000,000. All other analyses had sample sizes sufficiently large to reproduce computational/experimental values and report error estimates. |
| Data exclusions | Molecules for which no descriptor could be generated were excluded from the analyses. |
| Replication | Binding assays (Ki) were done in three independent replicates. Single-concentration radioligand displacements were done in two technical replicates. Computational experiments were done in three replicates where appropriate. All attempts at replication were successful. |
| Randomization | Random samples were extracted from chemical libraries that were shuffled using Terashuf. Class imbalance was maintained using scikit-learn's stratified split function. |
| Blinding | Blinding was not applicable to calculations in the study. Standardized experimental procedures did not necessitate blinding. |

# Reporting for specific materials, systems and methods

We require information from authors about some types of materials, experimental systems and methods used in many studies. Here, indicate whether each material, system or method listed is relevant to your study. If you are not sure if a list item applies to your research, read the appropriate section before selecting a response.

## Materials & experimental systems

| n/a | Involved in the study |
|---|---|
| ☒ | ☐ Antibodies |
| ☒ | ☐ Eukaryotic cell lines |
| ☒ | ☐ Palaeontology and archaeology |
| ☒ | ☐ Animals and other organisms |
| ☒ | ☐ Clinical data |
| ☒ | ☐ Dual use research of concern |
| ☒ | ☐ Plants |

## Methods

| n/a | Involved in the study |
|---|---|
| ☒ | ☐ ChIP-seq |
| ☒ | ☐ Flow cytometry |
| ☒ | ☐ MRI-based neuroimaging |

## Plants

| Seed stocks | N/A |
|---|---|
| Novel plant genotypes | N/A |
| Authentication | N/A |

