## [Peer Review File · Nature Computational Science]

Rapid Traversal of Vast Chemical Space using Machine Learning Guided Docking Screens

Corresponding Author: Professor Jens Carlsson

Version 0:

Decision Letter:

** Please ensure you delete the link to your author homepage in this e-mail if you wish to forward it to your co-authors. **

Dear Professor Carlsson,

Your manuscript "Rapid Traversal of Ultralarge Chemical Space using Machine Learning Guided Docking Screens" has now been seen by 4 referees, whose comments are appended below. You will see that while they find your work of interest, they have raised points that need to be addressed before we can make a decision on publication.

The referees' reports seem to be quite clear. Naturally, we will need you to address *all* of the points raised.

While we ask you to address all of the points raised, the following points need to be substantially worked on:

- 1) Please add additional discussion regarding the novelty of your approach compared to existing methods, while avoiding overhyped language.
- 2) Please update the references based on the recommendations from the reviewers.

Please use the following link to submit your revised manuscript and a point-by-point response to the referees' comments (which should be in a separate document to any cover letter):

Link Redacted

** This url links to your confidential homepage and associated information about manuscripts you may have submitted or be reviewing for us. If you wish to forward this e-mail to co-authors, please delete this link to your homepage first. **

To aid in the review process, we would appreciate it if you could also provide a copy of your manuscript files that indicates your revisions by making use of Track Changes or similar mark-up tools. Please also ensure that all correspondence is marked with your Nature Computational Science reference number in the subject line.

In addition, please make sure to upload a Word Document or LaTeX version of your text, to assist us in the editorial stage.

To improve transparency in authorship, we request that all authors identified as 'corresponding author' on published papers create and link their Open Researcher and Contributor Identifier (ORCID) with their account on the Manuscript Tracking System (MTS), prior to acceptance. ORCID helps the scientific community achieve unambiguous attribution of all scholarly contributions. You can create and link your ORCID from the home page of the MTS by clicking on 'Modify my Springer Nature account'. For more information please visit www.springernature.com/orcid.

We hope to receive your revised paper within three weeks. If you cannot send it within this time, please let us know.

Best regards,

Kaitlin McCardle, PhD
Senior Editor
Nature Computational Science

Reviewers comments:

Reviewer #1 (Remarks to the Author):

Reviewer 1 and Reviewer 2 have worked together on this review.

Summary.

In this work, Luttens et al. confront the growing problem of efficiently exploring the rapidly expanding chemical space available for drug discovery. They present a workflow designed to screen ultra-large chemical libraries at a fraction of the cost of traditional virtual screening approaches, utilizing a combination of machine learning and molecular docking with DOCK 3.7 software. The approach involves the selection and docking of a random training set of molecules, used to establish a docking score threshold for labeling molecules as “virtual actives” or “inactives”. The authors use this data to train and evaluate the performance of several machine-learning (ML) classifiers, including the tree-based CatBoost model, a Deep Neural Network (DNN), and a large language model (RoBERTa), using either Morgan2 fingerprints or continuous data-driven descriptors (CDDD) as molecular descriptors. They subsequently employ a Mondrian conformal predictor (CP) framework to classify the remaining molecules in the ultra-large chemical libraries into one of four categories: “virtual active”, “virtual inactive”, “virtual active or inactive”, and “no assignment”. The advantage of the CP framework is in the ability of the user to specify a numerical significance threshold for the prediction error rate, allowing for a significant reduction in the number of molecules selected for further evaluation by molecular docking based on the confidence of the ML classifier. To establish the performance of their approach, the authors developed a benchmarking set by conducting a screen against eight protein targets using 11 million molecules obtained from filtering the Enamine REAL library. They used these screens to evaluate the performance of their classification models, identifying the CatBoost classifier with Morgan2 descriptor as the best performing and the least computationally expensive. Subsequently, this model was used for the screening of ultra-large (235 million) and multi-billion (3.5 billion) scale chemical libraries against the Adenosine-2A and Dopamine-D2 receptors. They further demonstrate their approach by screening for molecules exhibiting polypharmacology against both receptors. As a proof of concept, the authors ordered 45 molecules identified through their workflow and performed experimental radioligand assays, obtaining lead candidates with micromolar binding constants against both A2A and Dopamine-D2 receptors. They found one molecule that bound to both proteins.

General comments.

This work represents a meaningful and significant contribution to the field of drug discovery and virtual screening using molecular docking, establishing an easily reproducible workflow for efficiently exploring the vast chemical space of billions of molecules currently available through make-on-demand chemical libraries, expected to continue to grow in the coming years. The work appears to be comprehensive, scientifically accurate, and technically sound. However, the significance of the implementation of the CP framework, especially for those not familiar with the approach, may be a bit difficult to follow without the reading of additional literature. Furthermore, in recent years, several papers in the literature (detailed below) have described and implemented “active learning” approaches similar to that presented in this work, although, to our knowledge, none have tested their predictions experimentally as is done here. Also, the polypharmacology aspect of this work is unique.

The authors have provided a codebase that is updated and maintained on GitHub, facilitating the reproducibility of the findings and data analysis performed in this paper.

Questions.

1. Page 9: The authors mention two types of molecular descriptors used for evaluating the CatBoost and DNN classifiers (Morgan2 fingerprints and CDDD). Is there any specific reason that these descriptors were chosen? Besides Morgan2, several other types of molecular fingerprints are available, e.g. ECFP4, MHFP6, and MAP4. Can the authors comment on whether their classifiers would perform differently if they used these different molecular representations?
2. Page 13: The authors aptly note that the selection of the protein and the docking parameters need to be properly optimized through enrichment calculations using small sets of actives and decoys. However, they do not provide the details of their enrichment calculations for the optimization of their workflow against the A2AR and D2R targets. Could the authors describe the process by which they performed enrichment calculations against these targets, and how they obtained actives and decoys to perform these calculations? Could they provide any LogAUC plots for these enrichment calculations?
3. Over the last few years, several papers have been published in the literature with “active learning” approaches that aim to reduce the computational cost of screening ultra-large chemical libraries, similar to the one presented in this work (e.g., doi: 10.1038/s41596-021-00659-2, 10.1021/acs.jctc.1c00810, 10.1039/d0sc06805e, and 10.1021/acs.jcim.3c01661). As in these other papers, the final step of this work involves selecting a subset of promising compounds that are eventually docked to the target and inspected. The primary difference presented in this paper only seems to be in the use of the conformal predictor framework, based on the outputs of multiple ML classification algorithms instead of a single deep neural network. Can the authors comment further on the novelty of their approach, or otherwise discuss the utility of their approach in the context of other similar approaches existing in the literature? They might consider stressing the ligand discovery aspect and the polypharmacology aspect.

4. Pages 6-7 (Steps 1-2 of pipeline): As presented, the eventual predictive abilities of the conformal predictor are heavily dependent on the docking score threshold labeling arising from the chosen training set, which is a randomly selected subset of a chemical library. There is an assumption that this random chemical subset exhibits docking scores that are sufficiently representative of chemical space to allow for training classifiers to make predictions about the remaining molecules in the ultra-large library (as the authors allude to on pg. 12). In this work, the authors use the Enamine REAL database to obtain a random sampling for their training set. However, it could be envisioned that companies interested in the approach presented here may want to use their own internal chemical libraries for this purpose, which may be biased or enriched towards a particular area of chemical space. Consequently, it is possible that docking score thresholds obtained from preparation and labeling of such a biased training set (even with multiple independently generated sets) may not generate classifiers with sufficient predictive power to categorize meaningful "virtual actives" in an ultra-large library. Could the authors comment on the robustness (i.e., predictive and reductive capability) of their method when given an initial "biased" training set? Do the authors have specific "best-practice" recommendations to avoid this situation from occurring?

5. In the same vein as point 4, "Data-dimensionality reduction (UMAP) indicates that these prioritized molecules bear structural similarity to the actives present in the training set ... supported by analysis of Tanimoto similarity." It sounds like the training set, the subset of the library docked that the authors then train on, impacts the resulting molecules selected. How should one choose this initial set of molecules? Should this choice of molecules be target/application dependent, or should you always train on the same set? "[T]his library was obtained after shuffling the SMILES with Terashuf", by this it sounds like the molecules are randomly selected and the same set are used for all targets. Is that right? Would being more mindful about choosing the molecules make a difference, for example choosing cluster heads or Synthons. This way more chemical diversity will be insured rather than depending on randomness. We would like the authors to comment on this.

6. Could the Authors discuss Synthons? Could this approach be incorporated into the method described here, could this complement this approach or should it remain a separate method. (doi:10.1038/s41586-021-04220-9).

7. What is the cost of training these machine learning models? How long does it take? How many machines does it run on? how much memory does it use ... etc. how does the cost vary with training size?

8. And after this training. How long does it take to screen 3.5 B molecules through your ML method?

9. After the machine learning model is trained what does the input output look like. It seems like you give it a SMILES, and it outputs two numbers: "A pair of p-values (p1 referring to the confidence the sample belongs to the virtual actives and p0 referring to the confidence the sample belongs to the virtual inactive class)". How do these numbers correlate with docking score? Could the authors show a correlation plot (score and rank) between dock and machine learning output. We would expect that there is some correlation between DOCK score and machine learning output. Is this true? Could the authors quantify this. Perhaps the authors could report Pearson and/or Spearman correlation coefficients.

10. Could the authors discuss the bookkeeping needed to dock these molecules after classification. It seems that the authors downloaded the ZINC database, both smiles and db2 files (compressed using gzip and tar). They pass the smiles through their machine learning method to get a prioritized list. They then go to the ZINC downloaded database and collate the db2 hierarchy associated with each molecule. Is this right? Or, are we getting this wrong? It might be helpful to outline the workflow in the methods. Upon rereading the methods, it seems that the authors did not download ZINC but built the molecules (Enamine Real) themselves. Did they just build the molecules that they are docking? If so, how long did this building take? The authors might consider clarifying this.

11. Although the authors show binding by radioligand displacement, we feel that there is not much discussion of the biological behavior of the molecules discovered in this paper. Perhaps, this is beyond the scope of this paper, but we are interested in the biology of these molecules. It might be sufficient for the authors to state if they are pursuing these molecules with further studies to characterize their biological behavior.

a. Are the authors pursuing these molecules to develop them further by exploring their biological behavior?

b. Are the molecules agonists or antagonists?

c. We are particularly interested in knowing more for the polypharmacological molecule. The authors seem to want to discover a molecule that is an agonist to D2R and an antagonist to A2AR.

d. Why were not all of the molecules tested in the D2R assay?

e. Why is the poly pharmacology aspect not stressed more in the abstract or conclusion? How are the authors developing the hit that showed binding to both A2AR and D2R. Does this molecule bind to more GPCRs?

Minor comments:

12. In the introduction, the authors discuss QSAR but do not fully relate their method to QSAR methods in the rest of the document. Could the authors put more into context how their methods are QSAR methods. For example, it seems that the continuous data-driven descriptors (CDDD) is a QSAR method. Is that right? Maybe state it in the methods, just to tie in this

introduction paragraph into the rest of the document. It seemed isolated and unclear the described methods were QSAR methods.

13. Figure 3: The figure descriptions for panels (a), (b), (c), and (d) seem to be scrambled, while panels (e) and (f) are accurately described. It seems that (a) describes panel (d), (b) describes panel (a), (c) describes panel (b), and (d) describes panel (c). To avoid confusion, please correct the correspondence between panel description and label.

14. Figure 4: Could the authors provide docking scores for molecules 1 and 2 shown in panel (b)? This would allow the reader to gain an appreciation for where these molecules fall on the distribution shown in panel (a).

15. For Figure 4 A. There appear to be four curves (Blue, light blue, pink, red) not just one in blue as is stated in the figure caption. It is also hard to see the differences between these four plots as they are on top of one another, particularly for the left side of the curve. We assume that the colors correspond to those discussed in Figure 3B where colors represent different significance thresholds.

16. Text in some of the figures is too small to read, e.g. Figure 5 panels B and C. We would suggest making the font bigger in the legend for these two panels.

17. We see in the supplemental document, Supplemental Table 1, the authors provide which residues are polarized (targed), but it is not clear how they redistributed the partial charge. Could the author describe the polarization of residues (targing)? Specifically, for which atoms are the partial charges altered?

Reviewer #2 (Remarks to the Author):

This manuscript was co-reviewed with Reviewer 1.

Summary.

In this work, Luttens et al. confront the growing problem of efficiently exploring the rapidly expanding chemical space available for drug discovery. They present a workflow designed to screen ultra-large chemical libraries at a fraction of the cost of traditional virtual screening approaches, utilizing a combination of machine learning and molecular docking with DOCK 3.7 software. The approach involves the selection and docking of a random training set of molecules, used to establish a docking score threshold for labeling molecules as “virtual actives” or “inactives”. The authors use this data to train and evaluate the performance of several machine-learning (ML) classifiers, including the tree-based CatBoost model, a Deep Neural Network (DNN), and a large language model (RoBERTa), using either Morgan2 fingerprints or continuous data-driven descriptors (CDDD) as molecular descriptors. They subsequently employ a Mondrian conformal predictor (CP) framework to classify the remaining molecules in the ultra-large chemical libraries into one of four categories: “virtual active”, “virtual inactive”, “virtual active or inactive”, and “no assignment”. The advantage of the CP framework is in the ability of the user to specify a numerical significance threshold for the prediction error rate, allowing for a significant reduction in the number of molecules selected for further evaluation by molecular docking based on the confidence of the ML classifier. To establish the performance of their approach, the authors developed a benchmarking set by conducting a screen against eight protein targets using 11 million molecules obtained from filtering the Enamine REAL library. They used these screens to evaluate the performance of their classification models, identifying the CatBoost classifier with Morgan2 descriptor as the best performing and the least computationally expensive. Subsequently, this model was used for the screening of ultra-large (235 million) and multi-billion (3.5 billion) scale chemical libraries against the Adenosine-2A and Dopamine-D2 receptors. They further demonstrate their approach by screening for molecules exhibiting polypharmacology against both receptors. As a proof of concept, the authors ordered 45 molecules identified through their workflow and performed experimental radioligand assays, obtaining lead candidates with micromolar binding constants against both A2A and Dopamine-D2 receptors. They found one molecule that bound to both proteins.

General comments.

This work represents a meaningful and significant contribution to the field of drug discovery and virtual screening using molecular docking, establishing an easily reproducible workflow for efficiently exploring the vast chemical space of billions of molecules currently available through make-on-demand chemical libraries, expected to continue to grow in the coming years. The work appears to be comprehensive, scientifically accurate, and technically sound. However, the significance of the implementation of the CP framework, especially for those not familiar with the approach, may be a bit difficult to follow without the reading of additional literature. Furthermore, in recent years, several papers in the literature (detailed below) have described and implemented “active learning” approaches similar to that presented in this work, although, to our knowledge, none have tested their predictions experimentally as is done here. Also, the polypharmacology aspect of this work is unique.

The authors have provided a codebase that is updated and maintained on GitHub, facilitating the reproducibility of the findings and data analysis performed in this paper.

Questions.

1. Page 9: The authors mention two types of molecular descriptors used for evaluating the CatBoost and DNN classifiers (Morgan2 fingerprints and CDDD). Is there any specific reason that these descriptors were chosen? Besides Morgan2,

several other types of molecular fingerprints are available, e.g. ECFP4, MHFP6, and MAP4. Can the authors comment on whether their classifiers would perform differently if they used these different molecular representations?

2. Page 13: The authors aptly note that the selection of the protein and the docking parameters need to be properly optimized through enrichment calculations using small sets of actives and decoys. However, they do not provide the details of their enrichment calculations for the optimization of their workflow against the A2AR and D2R targets. Could the authors describe the process by which they performed enrichment calculations against these targets, and how they obtained actives and decoys to perform these calculations? Could they provide any LogAUC plots for these enrichment calculations?

3. Over the last few years, several papers have been published in the literature with “active learning” approaches that aim to reduce the computational cost of screening ultra-large chemical libraries, similar to the one presented in this work (e.g., doi: 10.1038/s41596-021-00659-2, 10.1021/acs.jctc.1c00810, 10.1039/d0sc06805e, and 10.1021/acs.jcim.3c01661). As in these other papers, the final step of this work involves selecting a subset of promising compounds that are eventually docked to the target and inspected. The primary difference presented in this paper only seems to be in the use of the conformal predictor framework, based on the outputs of multiple ML classification algorithms instead of a single deep neural network. Can the authors comment further on the novelty of their approach, or otherwise discuss the utility of their approach in the context of other similar approaches existing in the literature? They might consider stressing the ligand discovery aspect and the polypharmacology aspect.

4. Pages 6-7 (Steps 1-2 of pipeline): As presented, the eventual predictive abilities of the conformal predictor are heavily dependent on the docking score threshold labeling arising from the chosen training set, which is a randomly selected subset of a chemical library. There is an assumption that this random chemical subset exhibits docking scores that are sufficiently representative of chemical space to allow for training classifiers to make predictions about the remaining molecules in the ultra-large library (as the authors allude to on pg. 12). In this work, the authors use the Enamine REAL database to obtain a random sampling for their training set. However, it could be envisioned that companies interested in the approach presented here may want to use their own internal chemical libraries for this purpose, which may be biased or enriched towards a particular area of chemical space. Consequently, it is possible that docking score thresholds obtained from preparation and labeling of such a biased training set (even with multiple independently generated sets) may not generate classifiers with sufficient predictive power to categorize meaningful “virtual actives” in an ultra-large library. Could the authors comment on the robustness (i.e., predictive and reductive capability) of their method when given an initial “biased” training set? Do the authors have specific “best-practice” recommendations to avoid this situation from occurring?

5. In the same vein as point 4, “Data-dimensionality reduction (UMAP) indicates that these prioritized molecules bear structural similarity to the actives present in the training set ... supported by analysis of Tanimoto similarity.” It sounds like the training set, the subset of the library docked that the authors then train on, impacts the resulting molecules selected. How should one choose this initial set of molecules? Should this choice of molecules be target/application dependent, or should you always train on the same set? “[T]his library was obtained after shuffling the SMILES with Terashuf”, by this it sounds like the molecules are randomly selected and the same set are used for all targets. Is that right? Would being more mindful about choosing the molecules make a difference, for example, choosing cluster heads or Synthons? This way, more chemical diversity will be ensured, rather than depending on randomness. We would like the authors to comment on this.

6. Could the authors discuss Synthons? Could this approach be incorporated into the method described here? Could this complement this approach, or should it remain a separate method? (doi:10.1038/s41586-021-04220-9).

7. What is the cost of training these machine learning models? How long does it take? How many machines does it run on? How much memory does it use ... etc. How does the cost vary with training size?

8. After this training, how long does it take to screen 3.5 B molecules through your ML method?

9. After the machine learning model is trained, what does the input output look like? It seems like you give it a SMILES, and it outputs two numbers: “A pair of p-values (p1 referring to the confidence the sample belongs to the virtual actives and p0 referring to the confidence the sample belongs to the virtual inactive class)”. How do these numbers correlate with docking score? Could the authors show a correlation plot (score and rank) between dock and machine learning output? We would expect that there is some correlation between DOCK score and machine learning output. Is this true? Could the authors quantify this? Perhaps, the authors could report Pearson and/or Spearman correlation coefficients.

10. Could the authors discuss the bookkeeping needed to dock these molecules after classification? It seems that the authors downloaded the ZINC database, both smiles and db2 files (compressed using gzip and tar). They pass the SMILES through their machine learning method to get a prioritized list. They then go to the ZINC downloaded database and collate the db2 hierarchy associated with each molecule. Is this right? Or are we getting this wrong? It might be helpful to outline the workflow in the methods. Upon rereading the methods, it seems that the authors did not download ZINC but built the molecules (Enamine Real) themselves. Did they just build the molecules that they are docking? If so, how long did this building take? The authors might consider clarifying this.

11. Although the authors show binding by radioligand displacement, we feel that there is not much discussion of the biological behavior of the molecules discovered in this paper. Perhaps, this is beyond the scope of this paper, but we are interested in the biology of these molecules. It might be sufficient for the authors to state if they are pursuing these molecules with further studies to characterize their biological behavior.

- a. Are the authors pursuing these molecules to develop them further by exploring their biological behavior?
- b. Are the molecules agonists or antagonists?
- c. We are particularly interested in knowing more for the polypharmacological molecule. The authors seem to want to discover a molecule that is an agonist to D2R and an antagonist to A2AR.
- d. Why were all the molecules not tested in the D2R assay?
- e. Why is the polypharmacology aspect not stressed more in the abstract or conclusion? How are the authors developing the hit that showed binding to both A2AR and D2R? Does this molecule bind to more GPCRs?

Minor comments:

12. In the introduction, the authors discuss QSAR, but do not fully relate their method to QSAR methods in the rest of the document. Could the authors put more into context how their methods are QSAR methods? For example, it seems that the continuous data-driven descriptor (CDDD) is a QSAR method. Is that right? Maybe state it in the methods, just to tie in this introduction paragraph into the rest of the document. It seemed isolated and unclear that the described methods were QSAR methods.

13. Figure 3: The figure descriptions for panels (a), (b), (c), and (d) seem to be scrambled, while panels (e) and (f) are accurately described. It seems that (a) describes panel (d), (b) describes panel (a), (c) describes panel (b), and (d) describes panel (c). To avoid confusion, please correct the correspondence between panel description and label.

14. Figure 4: Could the authors provide docking scores for molecules 1 and 2 shown in panel (b)? This would allow the reader to gain an appreciation for where these molecules fall on the distribution shown in panel (a).

15. For Figure 4A, there appear to be four curves (blue, light blue, pink, red), not just one in blue, as is stated in the figure caption. It is also hard to see the differences between these four plots as they are on top of one another, particularly for the left side of the curve. We assume that the colors correspond to those discussed in Figure 3B, where colors represent different significance thresholds.

16. Text in some of the figures is too small to read, e.g., Figure 5, panels B and C. We would suggest making the font bigger in the legend for these two panels.

17. We see in the supplemental document, Supplemental Table 1, that the authors provide which residues are polarized (targed), but it is not clear how they redistributed the partial charge. Could the authors describe the polarization of residues (targing)? Specifically, for which atoms are the partial charges altered?

Remarks on code availability.

The authors have released their code on GitHub (<https://github.com/carlssonlab/conformalpredictor>) with documentation and example usage.

Reviewer #2 (Remarks on code availability):

I have read through the README file provided on the GitHub repository that the authors have made publicly accessible, and successfully followed the instructions to install the codebase in a local environment. Moreover, I have looked through some of the Python scripts in the codebase, out of interest in understanding the practical implementation of the approach. The codebase provides a valuable resource to members of the scientific community interested in implementing this approach for drug discovery, molecular docking, and virtual screening against their own targets of interest.

Reviewer #3 (Remarks to the Author):

With their conformal prediction framework, the authors present another interesting approach to add to the rapidly evolving field of ML-accelerated ultralarge docking concepts. The manuscript is overall very well written and basically ready for publication, although I have some minor remarks that could improve it even further:

- Since the authors state in several places that their CP framework runs with modest resources, I feel that a bit more space should be devoted to discussing where the AMCP ranks in terms of the computational expense in comparison to other recent approaches, e.g., the different deep learning approaches in ref. 10-12, but also other recently revisited, more classical, ML approaches making similar claims (e.g., DOI 10.1021/acs.jcim.3c01661).

What architecture was used to perform this study, how much time and resources were spent for training and prediction steps? How do you anticipate your approach to scale with even larger libraries?

- Since some readers might be interested in this as a way to speed up VS without being familiar with CPs, the authors might want to dedicate a half-sentence to explaining what a Mondrian CP is (i.e., mention that this CP is for working with class

imbalance on the first mention).

- Page 11: Fix reference 28 in this paragraph for consistency.

- Page 15: It is of no major consequence, but the authors should clarify if they evaluate 3-5% of the full dataset of 235 million, or if they refer to 3-5% of the 234 million that were not used for training.

- Fig. 3b: The numbers in the significance dial inset are partly very hard to read with black font on top of dark red and blue. Can you change the font color to white in at least the two darkest slices?

- Figs. 4b and 5ef: Since you are showing docking results as opposed to experimental structures, displaying hydrogens would make this more 'intuitive' to note for a reader. The figure captions should also tell the reader what they are looking at - something along the lines of: cartoon representation of D2R with key interacting residues in sticks representation, hydrogen bonding indicated by dashed orange line.

Since the system does not tell whether comments on the code will be sent to the authors, I will also add some feedback on the code here:

- I would recommend changing the deprecated sklearn in your setup.py to scikit-learn. As is, your code will otherwise cause an error during installation on relatively recent OS (and this has a really simple fix):

```
× python setup.py egg_info did not run successfully.  
  | exit code: 1  
  └─> [15 lines of output]  
The 'sklearn' PyPI package is deprecated, use 'scikit-learn'  
rather than 'sklearn' for pip commands.
```

After changing that, I was able to install and run your code without any problems on my own data, which is unfortunately often enough not the case - thus, thank you for that.

However, I would recommend that you add a simple example run to your README that really just lists the necessary commands to do preparation, run training and prediction. While your current README explains the key options and tweaks for each step, I found myself getting a bit lost when trying to find the commands to run a quick example, just to see whether the code works or not, so the most basic test run in the beginning of the README would really elevate things for me.

Reviewer #3 (Remarks on code availability):

I was able to install the code on Linux after fixing a minor deprecation problem in setup.py (as pointed out to the authors):

```
× python setup.py egg_info did not run successfully.  
  | exit code: 1  
  └─> [15 lines of output]  
The 'sklearn' PyPI package is deprecated, use 'scikit-learn'  
rather than 'sklearn' for pip commands.
```

With that minor correction, I was able to complete the installation and run the code as instructed by the authors. I successfully used my own data, so it can be directly applied by the community.

I did not exhaustively test all available options, but the basic preparation steps, training and prediction steps completed without issues.

The README file is detailed and has a lot of explanation, although I missed a simple exemplary summary of which commands to run in which order for an easy start/test of setup.

The authors also provide their data for download on Zenodo, and while I have not verified that I get the same results, this should make the original study results reproducible, if desired.

Reviewer #4 (Remarks to the Author):

As chemical space becomes increasingly synthetically tractable, strategies for efficient virtual screening become increasingly valuable. In Lutten et al., they propose a principled method to use machine learning to define an efficient proxy score for a physics based docking energy and use it as a pre-screening filter to accelerate virtual screening. Their ML method extends the use of conformal prediction to large-scale docking, which gives a principled estimate of uncertainty. They conduct relevant benchmarking of their method assessing the role of molecular fingerprints, gradient boosted decision tree and deep-learning based ML method, and key method parameters including the model ensemble size, number of docked molecules, score-threshold, and sensitivity to noise. To test their method they apply it to 8 retrospective diverse and protein drug targets, retrospective out-of-distribution testing of experimentally tested compounds curated from the literature, and two prospective tests where they acquired 31 and 45 compounds and tested for IC50/binding activity, While they did find

active compounds, the results are modest but not cherry picked giving a credible assessment of the performance. In particular, for the second task, where they aim to identify polypharmacology for D2R and A2AR--which think this is an interesting test--but their discovered performance is quite weak (> 10 uM IC50). Overall The manuscript text and figures are clear, and the conclusions and claims are overall well supported.

Given the overall strength of the work, there are a few limitations that could be more clearly highlighted.

First, while they do cite prior work towards the problem of accelerating physics based virtual screening with ML, they there are at least 9 additional works that are highly comparable that they do not cite:

- * "Spresso: an ultrafast compound pre-screening method based on compound decomposition" (Yanagisawa, et al., Bioinformatics, 2017)
- * "State of the Art Iterative Docking with Logistic Regression and Morgan Fingerprints" (Martin, 2021, 10.26434/chemrxiv.14348117.v1)
- * "Lean-Docking: Exploiting Ligands' Predicted Docking Scores to Accelerate Molecular Docking" (Berenger, et al., 2021, 10.1021/acs.jcim.0c01452)
- * "Machine Learning Boosted Docking (HASTEN): An Open-source Tool To Accelerate Structure-based Virtual Screening Campaigns", (Kalliokoski, et al., 2021, MolInfo)
- * "Deep Surrogate Docking: Accelerating Automated Drug Discovery with Graph Neural Networks", (Hosseini, et al., Neurips 2022)
- * "Uni-Dock: GPU-Accelerated Docking Enables Ultralarge Virtual Screening" (Yu, et al., 2023, JCTC, 10.1021/acs.jctc.2c01145)
- * "Machine learning-boosted docking enables the efficient structure-based virtual screening of giga-scale enumerated chemical libraries", (Sivula, et al., JCIIM 2023)
- * "Hit Discovery using docking ENriched by GEnerative Modeling (HIDDEN GEM): A novel computational workflow for accelerated virtual screening of ultra-large chemical libraries" (Popov, et al., 2024, 10.1002/minf.202300207)
- * "Accelerating Molecular Docking using Machine Learning Methods" (Bande, et al., 2024, 10.1002/minf.202300167)

I would like to better understand in what ways their method is superior to these others, e.g. in terms of performance, computational cost, robustness etc.

Second, their use of conformal prediction is good, however they do not adequately explain why prior application of conformal prediction to learn chemoinformatic tasks does not naturally extend to their use case. Specifically from my reading of (Svensson et al., 2017) they also used very similar molecule encodings (RDKit descriptors vs. Morgan2) and ML based methods (Random Forest vs. CatBoost), but claim "Strategies to improve the virtual screening efficiency using the CP framework have been explored, but these workflows were not suitable for multi-billion-scale libraries and focused on traditional classifiers.", why is this? Additionally, I would appreciate it if the authors could more clearly articulate the advantages of the conformal prediction as many readers may not be familiar with it.

Third, the authors should address why the predictive accuracy is less than what was found e.g. for D4R using a highly physics based virtual screening method in (Lyu et al., 2021). Specifically, does the proxy ML model hurt down performance? One concern is that ML based proxy methods may be focusing on the common chemotypes and ignoring rare chemotypes, effectively decreasing the diversity of predicted ligands, thus this would increase the similarity of selected hits and thus increase the variance in discovery. This can be directly measured e.g. through quantifying the diversity of the predicted vs. docking hits. Note this lack of diversity is partially apparent in the UMAP panel 3E the blue "Priority" points are much more concentrated than the red "Active train set" points.

Specific textual comments:

Page 3:

In referencing the size of accessible chemical space, consider citing BioSolveIT's KnowledgeSpace, as they claim they are enumerating 290,000,000,000,000 drug-like compounds.

(Bellmann, 2022, 10.1021/acs.jcim.2c00334)

Calculating and Optimizing Physicochemical Property Distributions of Large Combinatorial Fragment Spaces

"Therefore, there is an urgent need for more efficient virtual screening approaches able to evaluate multi-billion-scale libraries." => I think this needs justification, as it's not obvious that larger compound libraries are better. You could cite "Modeling the expansion of virtual screening libraries", (Lyu et al., NatChemBio, 2023, DOI: 10.1038/s41589-022-01234-w), which I think is relevant and you cite later in the manuscript for a different claim.

"Traditionally, QSAR models have been trained on experimental data, but there is an increasing interest to predict which compounds in make-on-demand libraries are likely to receive favorable scores from computationally expensive virtual screening methods." => You should also cite these additional methods for accelerating large-scale virtual screens:

"Spresso: an ultrafast compound pre-screening method based on compound decomposition" (Yanagisawa, et al., Bioinformatics, 2017)

"State of the Art Iterative Docking with Logistic Regression and Morgan Fingerprints" (Martin, 2021, 10.26434/chemrxiv.14348117.v1)

"Lean-Docking: Exploiting Ligands' Predicted Docking Scores to Accelerate Molecular Docking" (Berenger, et al., 2021,

10.1021/acs.jcim.0c01452)

"Machine Learning Boosted Docking (HASTEN): An Open-source Tool To Accelerate Structure-based Virtual Screening Campaigns", (Kalliokoski, et al., 2021, MolInfo)

"Deep Surrogate Docking: Accelerating Automated Drug Discovery with Graph Neural Networks", (Hosseini, et al., Neurips 2022)

"Uni-Dock: GPU-Accelerated Docking Enables Ultralarge Virtual Screening" (Yu, et al., 2023, JCTC, 10.1021/acs.jctc.2c01145)

"Machine learning-boosted docking enables the efficient structure-based virtual screening of giga-scale enumerated chemical libraries", (Sivula, et al., JCI 2023)

"Hit Discovery using docking ENriched by GEnerative Modeling (HIDDEN GEM): A novel computational workflow for accelerated virtual screening of ultra-large chemical libraries" (Popov, et al., 2024, 10.1002/minf.202300207)

"Accelerating Molecular Docking using Machine Learning Methods" (Bande, et al., 2024, 10.1002/minf.202300167)

For citing conformal prediction, cite the original method and not just tutorials about the method.

"Strategies to improve the virtual screening efficiency using the CP framework have been explored, but these workflows were not suitable for multi-billion-scale libraries and focused on traditional classifiers." => can you please say more about what the limitations of these methods were? From my reading of the e.g. of (Svensson et al., 2017) they also used very similar molecule encodings (RDKit descriptors vs. Morgan2) and ML based methods (Random Forest vs. CatBoost).

Reviewer #4 (Remarks on code availability):

I have run the code and it is easy to use it works. The simplicity of the method and the large-scale docking results across the targets is a great resource for the community as a future baseline method and test set.

Version 1:

Decision Letter:

Our ref: NATCOMPUTSCI-24-1139A

2nd January 2025

Dear Dr. Carlsson,

Thank you for submitting your revised manuscript "Rapid Traversal of Ultralarge Chemical Space using Machine Learning Guided Docking Screens" (NATCOMPUTSCI-24-1139A). It has now been seen by the original referees and their comments are below. The reviewers find that the paper has improved in revision, and therefore we'll be happy in principle to publish it in Nature Computational Science, pending minor revisions to satisfy the referees' final requests and to comply with our editorial and formatting guidelines.

TRANSPARENT PEER REVIEW

Nature Computational Science offers a transparent peer review option for original research manuscripts. We encourage increased transparency in peer review by publishing the reviewer comments, author rebuttal letters and editorial decision letters if the authors agree. Such peer review material is made available as a supplementary peer review file. **Please remember to choose, using the manuscript system, whether or not you want to participate in transparent peer review.**

Please note: we allow redactions to authors' rebuttal and reviewer comments in the interest of confidentiality. If you are concerned about the release of confidential data, please let us know specifically what information you would like to have removed. Please note that we cannot incorporate redactions for any other reasons. Reviewer names will be published in the peer review files if the reviewer signed the comments to authors, or if reviewers explicitly agree to release their name. For more information, please refer to our <https://www.nature.com/documents/nr-transparent-peer-review.pdf> target="new">FAQ page.

Thank you again for your interest in Nature Computational Science. Please do not hesitate to contact me if you have any questions.

Sincerely,

Kaitlin McCardle, PhD
Senior Editor
Nature Computational Science

ORCID

Reviewer #1 (Remarks to the Author):

Reviewer 1 and Reviewer 2 have worked together on this review.

We feel that the authors have addressed all our comments and the paper should be published.

Reviewer #1 (Remarks on code availability):

We looked at the code; please the comments from Reviewer 2.

Reviewer #2 (Remarks to the Author):

Reviewer 1 and Reviewer 2 have worked together on this review.

We have read through the responses to the questions that we posed in our original evaluation of this work, and looked at the additional figures and text that were added by the authors in response to all reviewers. We are satisfied with the revised manuscript, and feel that it is now acceptable for publication.

Following further evaluation of the code and its accompanying textual descriptions, we wanted to suggest one point where further clarification of the code would be appreciated (see "Remarks on code availability"). We would appreciate if the authors either added a comment in the codebase, or clarified the relevant sentence in the text.

Reviewer #2 (Remarks on code availability):

I have further assessed the code and the accompanying text descriptions provided in the manuscript. In response to other reviewers, the authors have added additional descriptions of the workflow in their README file, allowing new users to quickly implement the approach on their own datasets. I was able to successfully install the code on a HPC environment.

There is one aspect of the code where I think that further clarification would be appreciated:

1. The authors mention the following in their methods section: "The scikit-learn 0.24.2 package was used to perform a stratified split the datasets in proper training sets (80% of training set), calibration sets (20% of training set), and test sets. The ratio between virtual actives and inactives was maintained in all sets. This procedure was repeated using different random seeds to obtain independent sets."

From the codebase (`conformalpredictor/amcp/modes.py`), while it is evident that the authors indeed perform a stratified split of the data during training to obtain independent sets with proper ratios, the distinction between "calibration sets" and "test sets" is not very clear. By specifying `test_size=args.ratioTestSet` within the splitting argument, and assigning `y_calibration_data` to this value, it seems that the "calibration data" is actually considered the "test set", with no other explicit splitting of the dataset as a whole into a held-out "test set". Later in the code, within the "validation" method, the same `y_calibration_data` is assigned to `test_size=prop_test_ratio`, which is equivalent to `args.ratioTestSet`. Within the README file example, the authors mention that a separate file called `test_smiles.txt` should be specified to the program when making predictions, implying that the actual held-out "test set" splitting is not being done by the program itself, but by the user when they desire to make predictions on some test data. This aspect should be clarified to the user, so that they are aware that the stratified split is only being performed on the training data [which is being further split into "training (80%)" and "validation (20%)" (calibration) sets during K-fold cross-validation], and not the dataset as a whole (which would also include the held-out test data on which predictions are being performed).

Reviewer #3 (Remarks to the Author):

I have no further remarks on the revised version. I would simply like to thank the authors for their detailed responses to my previous comments and the implemented changes.

Reviewer #3 (Remarks on code availability):

I have only reviewed the changes made by the authors since the last test. The code already ran beforehand and the updated instructions match my notes.

Version 2:

Decision Letter:

Dear Professor Carlsson,

We are pleased to inform you that your Article "Rapid Traversal of Vast Chemical Space using Machine Learning Guided Docking Screens" has now been accepted for publication in Nature Computational Science.

Once your manuscript is typeset, you will receive an email with a link to choose the appropriate publishing options for your paper and our Author Services team will be in touch regarding any additional information that may be required.

Acceptance of your manuscript is conditional on all authors' agreement with our publication policies (see <https://www.nature.com/natcomputsci/for-authors>). In particular your manuscript must not be published elsewhere and there must be no announcement of the work to any media outlet until the publication date (the day on which it is uploaded onto our web site).

Before your manuscript is typeset, we will edit the text to ensure it is intelligible to our wide readership and conforms to house style. We look particularly carefully at the titles of all papers to ensure that they are relatively brief and understandable.

Once your manuscript is typeset, you will receive a link to your electronic proof via email with a request to make any corrections within 48 hours. If, when you receive your proof, you cannot meet this deadline, please inform us at rjsproduction@springernature.com immediately.

If you have queries at any point during the production process then please contact the production team at rjsproduction@springernature.com.

We welcome the submission of potential cover material (including a short caption of around 40 words) related to your manuscript; suggestions should be sent to Nature Computational Science as electronic files (the image should be 300 dpi at 210 x 297 mm in either TIFF or JPEG format). We also welcome suggestions for the Hero Image, which appears at the top of our [home page](http://www.nature.com/natcomputsci); these should be 72 dpi at 1400 x 400 pixels in JPEG format. Please note that such pictures should be selected more for their aesthetic appeal than for their scientific content, and that colour images work better than black and white or grayscale images. Please do not try to design a cover with the Nature Computational Science logo etc., and please do not submit composites of images related to your work. I am sure you will understand that we cannot make any promise as to whether any of your suggestions might be selected for the cover of the journal.

Best regards,

Kaitlin McCardle, PhD
Senior Editor
Nature Computational Science

P.S. Click on the following link if you would like to recommend Nature Computational Science to your librarian: https://www.springernature.com/gp/librarians/recommend-to-your-library

** Visit the Springer Nature Editorial and Publishing website at www.springernature.com/editorial-and-publishing-jobs for more information about our career opportunities. If you have any questions please click here. **

UPPSALA
UNIVERSITET

SciLifeLab

November 10th, 2024

Dear Dr. McCardle,

Thank you for the invitation to submit a revised version of our manuscript (*“Rapid Traversal of Ultralarge Chemical Space using Machine Learning Guided Docking Screens”*, Manuscript ID: NATCOMPUTSCI-24-1139). We also want to thank the referees for taking the time to review our manuscript and the many constructive suggestions.

We have carefully considered all comments from the reviewers and performed both new calculations and experiments, leading to a substantially stronger manuscript. We have also rewritten several sections based on the reviewers’ comments and introduced a separate discussion section. We include detailed descriptions of the computational resources required to use our method and the advantages of our approach compared to currently available methods. Furthermore, we provide more detailed guidelines on how to apply our approach in a revised tutorial. Please find below point-by-point responses to the referees with their comments in **Blue**. Relevant changes to the manuscript have been marked in **yellow** to facilitate the review process.

Reviewers #1 and #2

These reviewers worked together and appreciated our work: *“This work represents a meaningful and significant contribution to the field of drug discovery and virtual screening using molecular docking, establishing an easily reproducible workflow for efficiently exploring the vast chemical space of billions of molecules currently available through make-on-demand chemical libraries, expected to continue to grow in the coming years”*. The reviewers supported publication of our manuscript after addressing their comments, and we respond to the questions below:

Major comments:

1. Reviewers: *“Page 9: The authors mention two types of molecular descriptors used for evaluating the CatBoost and DNN classifiers (Morgan2 fingerprints and CDDD). Is there any specific reason that these descriptors were chosen? Besides Morgan2, several other types of molecular fingerprints are available, e.g. ECFP4, MHFP6, and MAP4. Can the authors comment on whether their classifiers would perform differently if they used these different molecular representations?”*

The molecular fingerprints mentioned by the reviewers belong to the class of substructure-based methods (ECFP4 and MHFP6). In benchmarking studies for small molecules (drug-like compounds), the substructure-based fingerprints often perform best and ECFP ranks as one of the top-performing methods (ref. 31). This is the reason why we selected the Morgan2 fingerprints, which is the RDKit implementation of ECFP4 and

have become widely used in the cheminformatics community. Although we cannot exclude that other fingerprints can give different results, the virtual screening results of descriptors in this class are generally strongly correlated, which is demonstrated in Figure 3 of ref. 31. MAP4 is an interesting new fingerprint in this category, which was designed to perform well both on small and large molecules (<https://doi.org/10.1186/s13321-020-00445-4>). This fingerprint would be interesting to explore if our screening approach would be applied to prediction of larger molecules such as peptides, but this is out of the scope of the present study.

Rather than adding another method in the substructure class, we complemented the bit-wise representation with continuous- (CDDD) and transformer-based (RoBERTa) descriptors. CDDD and RoBERTa were included as these novel and widely-used classes of methods have shown advantages over traditional methods (e.g., ECFP4) in several evaluations of QSAR applications (ref. 32). Each descriptor type captures molecular information differently, and the choice of representation can influence the ability of the model to learn and generalize. There are also practical factors to consider with the use of each method, such as the time to generate the descriptors and storage requirements.

In summary, we decided to select three fundamentally different molecular representations in our study. We describe our criteria for selecting the three representations on pages 8 and 9 in the revised manuscript.

2. Reviewers: “The authors aptly note that the selection of the protein and the docking parameters need to be properly optimized through enrichment calculations using small sets of actives and decoys. However, they do not provide the details of their enrichment calculations for the optimization of their workflow against the A_{2A}R and D₂R targets. Could the authors describe the process by which they performed enrichment calculations against these targets, and how they obtained actives and decoys to perform these calculations? Could they provide any LogAUC plots for these enrichment calculations?”

We thank the reviewers for pointing out that this step was not clearly described. Prior to carrying out a prospective virtual screen, several docking parameters are optimized to enhance the performance of the scoring function, which is described in detail by Bender *et al.* (ref. 26).

In the case of the screen targeting both the A_{2A}R and D₂R, the enrichment calculations and optimization were performed in three steps. We first selected 10 antagonists and 25 agonists of the A_{2A}R and D₂R receptors from the ChEMBL database, respectively. In a second step, we generated property-matched decoys according to the procedure described by Mysinger *et al.* (ref. 58). In the last step, we docked the actives and decoys to the receptors structures and evaluated different docking parameters (e.g., the radius of electrostatic and desolvation spheres in DOCK3.7) to identify the optimal settings for ligand enrichment.

In the revised manuscript, we have included a more detailed description of the generation of the ligand-decoy control set and optimization (pages 8, 19, and 26). We have also included the Receiver Operating Characteristic (ROC) curves in the supporting information (Supplementary Figure S14).

3. Reviewers: *“Over the last few years, several papers have been published in the literature with “active learning” approaches that aim to reduce the computational cost of screening ultra-large chemical libraries, similar to the one presented in this work (e.g., doi: 10.1038/s41596-021-00659-2, 10.1021/acs.jctc.1c00810, 10.1039/d0sc06805e, and 10.1021/acs.jcim.3c01661). As in these other papers, the final step of this work involves selecting a subset of promising compounds that are eventually docked to the target and inspected. The primary difference presented in this paper only seems to be in the use of the conformal predictor framework, based on the outputs of multiple ML classification algorithms instead of a single deep neural network. Can the authors comment further on the novelty of their approach, or otherwise discuss the utility of their approach in the context of other similar approaches existing in the literature? They might consider stressing the ligand discovery aspect and the polypharmacology aspect.”*

We thank the reviewers for this question, and there are indeed several papers reporting methods that address the challenge of screening large chemical libraries. We present the novelty and advantages of our method compared to other approaches below.

(1) A robust framework: We develop novel strategies to use conformal prediction for large-scale virtual screening. Unlike most other machine learning models, a conformal predictor yields a set of two p-values: p_1 , the confidence that the molecule belongs to the class of virtual actives, and p_0 , the confidence that the molecule belongs to the class of virtual inactives. The conformal prediction framework is grounded in a mathematical proof that integrates p-values with a user-defined significance level (ϵ) to ensure a guaranteed prediction error rate, an assurance that cannot be provided by other available methods. In addition, we demonstrate that sorting the predictions according to the difference of these p-values ($p_1 - p_0$, also known as the quality of information metric) prioritizes molecules with substantially better docking scores. We demonstrate a correlation between docking rank and the quality of information (See question 9, Figure RL3), making this metric very valuable in the selection of smaller subsets of compounds. Consequently, a conformal predictor achieves equivalent or better performance in prioritizing relevant molecules in a single iteration than other methods that rely on resource-intensive active learning (e.g., as in refs. 13-15).

(2) Modest computational cost: A conformal predictor is a wisdom-of-the-crowd method that aggregates outputs by multiple machine learning classifiers. Despite the requirement to train independent machine learning models, the computational cost associated with training is almost negligible (see our answer to question 7 below). Furthermore, our analyses show that gradient-boosted trees can reach superior performances in comparison to more resource-intensive deep neural networks or large language models (e.g., as in refs. 13 and 14). Conformal predictors do not require GPU hardware for training or predicting (e.g., as in ref. 13), which enables scoring of one billion compounds on a standard multi-core desktop computer in less than 5 hours (see question 8 below).

Our comparison between different molecular representations revealed that machine learning models trained on Morgan2 fingerprints lead to similar or better performances as those trained on latent or transformer-based descriptors (e.g., as in refs. 32 and 50). From a practical point of view, bit-vectors are the only feasible type of descriptor that can store

trillions of molecules long-term due to the ease of compression. For these reasons, we were able to readily process more than 3.5 billion compounds with our conformal predictor framework, which is one of the largest libraries ever evaluated by machine-learning accelerated virtual screening.

(3) Open science and accessibility. To enable the research field to perform screens of multi-billion-scale libraries and further development of such methods, we share the following items with the community: (A) A set of hyperparameters that generalize well across the different protein targets and do not require parameter optimization for initial predictions, (B) Code on our GitHub (<https://github.com/carlssonlab/conformalpredictor>) and (C) Large datasets for benchmarking of methods. Multiple research groups (refs 13, 15, and 42) benchmark their methods using the same dataset generated by Lyu *et al.* (AmpC 96 million and D₄R 138 million compounds, ref. 5). While we recognize the need for established benchmarking datasets, we found that the machine learning performance was strongly dependent on the protein target. If the field focuses on a small number of targets and the same datasets, the developed methods may not generalize well across protein targets. To address this shortcoming, we publicly share docking scores of 235 million molecules against several targets (<https://doi.org/10.5281/zenodo.7953917>).

(4) Prospective ligand discovery. As the reviewers noted, our manuscript represents one of the few studies that both includes development of a new method and prospective screens for novel ligands of several drug targets. The molecules discovered in the screen against the D₂ dopamine receptor expand the repertoire of D₂ agonist chemotypes, which could catalyze drug discovery efforts against this important target. In addition, we explore novel strategies to identify compounds with pharmacological profiles (multi-target ligands) and test these predictions experimentally. Our results illustrate exciting opportunities in drug discovery enabled by combining the conformal prediction framework with large libraries. These aspects of our work extend beyond the scope of the previous work, providing a valuable addition to the field.

As suggested by the reviewers, we have put emphasis on the ligand discovery aspect of our work in the abstract and introduction (pages 2 and 18). We have also included our key contributions to the field in the discussion (page 23-24).

4. Reviewer: *“As presented, the eventual predictive abilities of the conformal predictor are heavily dependent on the docking score threshold labeling arising from the chosen training set, which is a randomly selected subset of a chemical library. There is an assumption that this random chemical subset exhibits docking scores that are sufficiently representative of chemical space to allow for training classifiers to make predictions about the remaining molecules in the ultra-large library (as the authors allude to on pg. 12). In this work, the authors use the Enamine REAL database to obtain a random sampling for their training set. However, it could be envisioned that companies interested in the approach presented here may want to use their own internal chemical libraries for this purpose, which may be biased or enriched towards a particular area of chemical space. Consequently, it is possible that docking score thresholds obtained from preparation and labeling of such a biased training set (even with multiple independently generated sets) may not generate classifiers with sufficient*

predictive power to categorize meaningful “virtual actives” in an ultra-large library. Could the authors comment on the robustness (i.e., predictive and reductive capability) of their method when given an initial “biased” training set? Do the authors have specific “best-practice” recommendations to avoid this situation from occurring?”

As the reviewers point out, the conformal prediction framework relies on the principle of exchangeability between the training and objective sets. This means that similar distributions of chemotypes need to be present in the training and objective sets. A recent paper by Bellmann *et al.* (<https://doi.org/10.1021/acs.jcim.1c01378>, ref. 4) reported that although the make-on-demand databases provided by several chemical vendors now contain billions of molecules, the overlap between these libraries is surprisingly small. To simulate the scenario described by the reviewers (where internal “biased libraries” are used for training), we trained machine learning models on a random sample of one million Ro4 molecules from the WuXi’s GalaXi database (1.4 billion compound library) and performed predictions on ten million Ro4 compounds from the Enamine REAL space (3.5 billion compound library) for which we had ground-truth docking scores for two targets (A₂A_R and D₂R). The results were then compared to our standard protocol, in which a conformal predictor was first trained on one million random molecules from the Enamine REAL space and then used to predict the ten million compounds that are exchangeable with the training set. In this analysis, apart from the difference in the training set used, all other molecular docking and machine learning parameters were identical.

We first analyzed the chemical similarity between the two training sets (WuXi and Enamine) and the test set (Enamine). In agreement with the analysis of Bellman, unsupervised data-dimensionality reduction (UMAP) of the Morgan2 fingerprints indicated that the molecules found in the Enamine training set bear greater structural similarity to the molecules in the Enamine test set than the molecules in the WuXi training set (Figure RL1a). These results indicated that the WuXi training set and Enamine test set would not be fully exchangeable.

Next, we assessed the performance of the conformal predictor trained on either the Enamine or Wuxi training set, followed by predictions on ten million random molecules that were docked to the A₂A_R and D₂R (Enamine test set). We then monitored the model’s sensitivity as a function of the significance value (ϵ). The sensitivity is defined as the ratio of the true positives and all positives. True positives were true active molecules correctly classified by the conformal predictor, *i.e.*, in the predicted virtual active and both sets, and all positives were all molecules with a score better than the threshold used to define the virtual actives. Although both classifiers achieved maximum sensitivity at the same significance ($\epsilon = 0.12$), the model trained on exchangeable data (Enamine training set) was able to retrieve 89% (A₂A_R) and 92% (D₂R) of the true actives, whereas the model trained on non-exchangeable data (WuXi training set) identified just 19% (A₂A_R) and 30% (D₂R) of the true actives (see below figure, Figure RL1b).

These results show that the exchangeability between training and objective sets is crucial. We therefore strongly recommend retrieving a random subset for training from the chemical space one wants to use to perform predictions. These new results have been

added on pages 11-12 of the revised manuscript and in the supplementary material (Supplementary Figure S9).

Figure RL1. (a) Two-dimensional unsupervised UMAP projection illustrates the chemical relationships in high-dimensional feature space between the WuXi training set, Enamine training set and Enamine test sets. (b) The difference in sensitivity values obtained from a conformal predictor model (five independent CatBoost classifiers trained on one million molecules) as a function of the significance value (ϵ).

5. Reviewer: *“In the same vein as point 4, “Data-dimensionality reduction (UMAP) indicates that these prioritized molecules bear structural similarity to the actives present in the training set ... supported by analysis of Tanimoto similarity.” It sounds like the training set, the subset of the library docked that the authors then train on, impacts the resulting molecules selected. How should one choose this initial set of molecules? Should this choice of molecules be target/application dependent, or should you always train on the same set? “[T]his library was obtained after shuffling the SMILES with Terashuf”, by this it sounds like the molecules are randomly selected and the same set are used for all targets. Is that right? Would being more mindful about choosing the molecules make a difference, for example choosing cluster heads or Synthons. This way more chemical diversity will be insured rather than depending on randomness. We would like the authors to comment on this.”*

These are important questions, which we address below:

• **Does the subset of the library docked influence the molecules selected?**

It is correct that we used a single (randomly selected) set of compounds for all targets. This is advantageous because we only need to prepare this set once and can use it for docking repeatedly, reducing computational costs. We agree that the initial selection of compounds to build a training set (a random sample) could impact the final molecules prioritized by the conformal predictor. However, if the training sets are exchangeable with the test and sufficiently large, the specific selection of molecules should not significantly impact the results. This is clear from our docking calculations of 235 million molecules to the A_{2A}R and D₂R. For each target, we divided the resulting datasets three times to obtain in one million random molecules for training and used the remaining 234 million molecules

for testing. The three independently trained conformal predictors yield very similar recall values of the 10000 top-ranked molecules (Figure 3d), demonstrating that a random selection of training set works very well and will lead to similar selection of molecules.

• **How should the initial set be selected? Cluster heads, synthons, target-based?**

As demonstrated in the previous question, it is crucial that the training set and the objective set are exchangeable. A biased training set (e.g., by clustering on molecular topology or synthons) would not be exchangeable with unbiased objective sets, and the conformal predictor would perform poorly, as shown in the previous question (we will discuss synthons more in detail in the following question). Finally, it is possible to introduce bias based on the target, e.g. based on the physicochemical properties of known ligands. For examples, one could consider to only screen cationic compounds for the D₂ dopamine receptor as most of the ligands are positively charged. In this case, the same filtering would need to be used for both the training and objective sets, leading to exchangeability and hence a valid conformal predictor.

We have mentioned the random selection of compounds on page 16 and discuss the exchangeability criterium by in depth on pages 11-12.

6. Reviewers: *“Could the Authors discuss Synthons? Could this approach be incorporated into the method described here, could this complement this approach or should it remain a separate method. (doi:10.1038/s41586-021-04220-9).”*

We thank the reviewers for this interesting question. Synthons represent a valuable concept that enables hierarchical navigation of vast chemical spaces, e.g. V-SYNTHES (ref. 43). In hierarchical approaches, small fragment-sized compounds in the database (synthons) are docked to the binding site. Top-scoring synthons are then identified and, in a second step, larger compounds containing these synthons are docked to the target protein. This divide-and-conquer approach is a very efficient technique to navigate large chemical libraries. However, integrating synthons into our machine learning workflow presents several challenges and considerations. In the first step of a hierarchical approach, there is no need for machine learning to identify the top-ranked synthons as the synthon libraries are relatively small (less than one million compounds). In the second step of the method, one potential strategy would be to use the synthon library as the training set and perform predictions in the enumerated library with billions of compounds. However, this is problematic for several reasons. Synthons represent small molecular fragments used in synthetic chemistry and will generally have worse scores than the larger product molecules from the enumerated library. Since the labelling of training data relies on a docking score threshold, synthons categorized as virtual actives are very unlikely to belong to the top-scoring minority class among the docked enumerated molecules. In other words, the conformal predictor would not perform well if the training is performed on docking of synthons (see previous point for an illustrative example).

For these reasons, the machine learning and synthon based methods should be considered complementary. An apple-to-apples comparison of the computational costs and ability of the methods to identify top-ranked compounds in large chemical libraries has not yet been presented and would be informative. We have added a discussion about the synthon-based approaches on pages 22-23 of the revised manuscript.

7. Reviewers: “What is the cost of training these machine learning models? How long does it take? How many machines does it run on? how much memory does it use ... etc. how does the cost vary with training size?”

The time required to train the conformal predictor models (consisting of five independent CatBoost classifiers) increases with training set size (see below figure, Figure RL2). Our recommended training set size is one million molecules. In this case, our method requires on average 1410 seconds to train a conformal predictor with no significant fluctuation between the protein targets. To train a conformal predictor on one million molecules, our method requires approximately 4 GB of RAM. The CPU-cores we had access to were 2x Intel Xeon Gold 6130 CPUs @ 2.10 GHz. Both training and prediction times increased significantly when using CDDD representations (Table 2). Although our models were trained on CPUs, training and predicting using the RoBERTa architecture benefits from GPU usage. For this, we used twelve cores of a single Nvidia Tesla T4 GPU with 1844 GiB memory. In this case, average training times amounted to 376685 seconds, which corresponds to approximately 21 GPU-hours per RoBERTa model.

Considering the comparable predictive power between CatBoost and RoBERTa models, CatBoost was selected because the cost of training was >250-fold lower. We have added a summary of computational costs and a description of the hardware employed in the methods (page 30) section of our revised manuscript.

Figure RL2. Required time (seconds) to train conformal predictors (consisting of five independent CatBoost classifiers) on training datasets of different sizes.

8. Reviewers: “And after this training. How long does it take to screen 3.5 B molecules through your ML method?”

To predict all molecules in the Rule-of-Four library (3,541,746,925 compounds), our method requires just over 115 CPU-hours (117 seconds per 1 million compounds). As we had access to a total of 500 cores, this process could be effectively parallelized to approximately 15 minutes wall-time. We have included this information in the methods section of the manuscript (page 30).

9. Reviewers: *“After the machine learning model is trained what does the input output look like. It seems like you give it a SMILES, and it outputs two numbers: “A pair of p-values (p1 referring to the confidence the sample belongs to the virtual actives and p0 referring to the confidence the sample belongs to the virtual inactive class)”. How do these numbers correlate with docking score? Could the authors show a correlation plot (score and rank) between dock and machine learning output. We would expect that there is some correlation between DOCK score and machine learning output. Is this true? Could the authors quantify this. Perhaps the authors could report Pearson and/or Spearman correlation coefficients.”*

We thank the reviewers for these interesting comments regarding how the machine learning predictions correlate with docking scores. In contrast to previously published methods that use regressors (e.g., ref. 13), our approach does not involve training the machine learning models to predict docking scores. After docking the training set, we divide the compounds into sets of virtual actives and inactives based on the docking scores. The models are then trained to identify the top-scoring 1% of the library. Our method takes the SMILES string of a molecule as input and returns two p-values as output: p_1 , (the confidence that the molecule belongs to the virtual actives class) and p_0 (the confidence that the molecule belongs to the virtual inactives class). As we did not train on or predict docking scores, we are not surprised to find that the correlations between the p-values (p_1 , p_0 , and their difference, $p_1 - p_0$) and docking scores were weak for (mean Pearson correlation coefficient and standard deviation: -0.097 ± 0.047).

Despite the lack of strong correlation with the docking scores, it is clear that difference between the p-values ($p_1 - p_0$, quality of information) is a useful metric to identify top-scoring compounds among the predicted virtual actives. If we instead used the docking rank in our analysis, we observe stronger correlations (mean of Pearson correlation coefficient and standard deviation: -0.518 ± 0.073) and the results for several targets are comparable to methods that trained their methods to predict docking scores (e.g., ref. 13). In the figure on the following page (Figure RL3) we show the how the distribution of the quality of information metric ($p_1 - p_0$) changes depending on the rank of the compounds for the eight protein targets. The separation is particularly clear for the top-ranked 1% of the library, which was the goal of the machine learning task. These results are consistent with Figure 3d, which shows how the quality of information metric can be used to prioritize compounds with better docking scores. In accordance with our results (Figure 2d-e), the correlation was strongest for the targets with lower optimal significance values and higher sensitivity values (e.g., AmpC, A_{2A}R, and D₂R), whereas targets with less well-behaving binding sites (e.g., SORT1, KEAP1, and M^{pro}) led to less distinct separations of the top-ranked 1%. We have included these new results in the revised manuscript (page 15) and Supplementary Figure S10.

Figure RL3. Correlations between the quality of information metric and molecular docking results. (a) Pearson correlation coefficients between the quality of information metric ($p_1 - p_0$) and molecular docking results (ranks / scores) for eight different protein targets. (b-i) For eight different protein targets, boxplots representing the distribution of the quality of information metric ($p_1 - p_0$) across different segments of the library (ten million molecules) ranked by docking scores. Each box spans from the first quartile (Q1, 25th percentile) to the third quartile (Q3, 75th percentile), with the purple line inside the box indicating the median (50th percentile). The whiskers extend to the most extreme data points within 1.5 times the interquartile range (IQR) from the quartiles. Data points outside this range are considered outliers and are not visualized for clarity. Blue and red dots respectively represent the median p_1 and p_0 values for the different segments.

10. Reviewers: *“Could the authors discuss the bookkeeping needed to dock these molecules after classification. It seems that the authors downloaded the ZINC database, both smiles and db2 files (compressed using gzip and tar). They pass the smiles through their machine learning method to get a prioritized list. They then go to the ZINC downloaded database and collate the db2 hierarchy associated with each molecule. Is this right? Or, are we getting this wrong? It might be helpful to outline the workflow in the methods. Upon rereading the methods, it seems that the authors did not download ZINC but built the molecules (Enamine Real) themselves. Did they just build the molecules that they are docking? If so, how long did this building take? The authors might consider clarifying this.”*

We thank the reviewers for this question. We actually both used the ZINC and Enamine libraries, but in different parts of the project. As correctly noted by the reviewers, we did not make use of the ZINC database for the docking and predictions of the Rule-of-Four datasets. Instead, we focused on molecules in the Enamine REAL databases (November 2019 and November 2022 versions). This facilitated our search for readily synthesizable compounds through a single chemical vendor. In-house preparation of these Rule-of-Four molecules approximately takes 18 CPU-seconds per compound. As a result, the one million molecules inside our training dataset were prepared for docking in approximately 5000 CPU-hours. Since we had access to 500 CPU cores, this task could be effectively parallelized to completion in 10 hours wall-time. Although the computational cost of generating conformational ensembles of molecules in ready-to-dock (db2) format may seem high, we would like to emphasize the following three items: (1) our machine learning method is agnostic of the scoring function and could thus also employ scoring functions with on-the-fly conformer generation, (2) once generated, the molecules in the training data can be reused in virtual screens against several targets, and (3) the relative cost of preparing molecules in this training dataset is insignificant compared to the cost of preparing the entire commercial chemical space for large-library docking. In order to generate large-scale datasets with ground-truth labels for validation of our conformal predictor framework, we explicitly docked a locally stored snapshot (235 million molecules) of the ZINC15 database against eight different targets. As suggested by the reviewers, we have included this information on page 14 and the method section of the revised manuscript.

11. Reviewers: *“Although the authors show binding by radioligand displacement, we feel that there is not much discussion of the biological behavior of the molecules discovered in this paper. Perhaps, this is beyond the scope of this paper, but we are interested in the biology of these molecules. It might be sufficient for the authors to state if they are pursuing these molecules with further studies to characterize their biological behavior.”*

We agree with the reviewers and have carried out additional experiments assays to further evaluate the properties of the discovered compounds.

Are the authors pursuing these molecules to develop them further by exploring their biological behavior?

The primary objective of this paper was to develop a robust machine learning platform for virtual screening. Encouraged by the strong performance in retrospective assessments, we extended our evaluation to include two prospective virtual screens. The successful identification of ligands in both screens serves as compelling proof-of-principle that our approach can effectively navigate vast chemical spaces to identify novel starting points for drug discovery. However, we did not perform hit-to-lead optimization for the dual-target ligand because this is a resource-intensive process and beyond the scope of this work.

Are the molecules agonists or antagonists? We are particularly interested in knowing more for the polypharmacological molecule. The authors seem to want to discover a molecule that is an agonist to D2R and an antagonist to A2AR.

We have performed additional functional experiments for compounds **1** and **2**, which were identified in the screen against the D₂R. Both ligands were full agonists with EC₅₀ values of 10 and 14 μM (E_{max} = 99% and 100%, respectively, relative to the maximal effect of dopamine). The potencies were hence 3- to 4-fold lower than the affinities. For this reason, the dual-target ligand (compound **5**) was not further evaluated in functional assays because of the modest affinity of this compound. Further optimization of the affinity of this scaffold would be required to characterize the biological behavior of this compound. We have included the new functional data on page 18 and in Supplementary Figure S13 of the revised manuscript.

Why were all the molecules not tested in the D2R assay?

The experimental evaluation of the predicted ligands from the polypharmacology screen was performed in two steps, and the A_{2A}R binding assay was the first of these. As the goal of the screen was to discover dual-target ligands, there was no reason to evaluate compounds at the D₂R that did not bind to the A_{2A}R.

Why is the polypharmacology aspect not stressed more in the abstract or conclusion? How are the authors developing the hit that showed binding to both A2AR and D2R? Does this molecule bind to more GPCRs?

We agree with the reviewers and have added a more detailed description of the prospective virtual screens in the abstract. Virtual screens to identify compounds with polypharmacological profiles is a very exciting application of machine-learning accelerated screens. We are currently identifying analogs of the dual-target ligand with the aim to further optimize the activity.

Minor comments:

12. Reviewers: *“In the introduction, the authors discuss QSAR but do not fully relate their method to QSAR methods in the rest of the document. Could the authors put more into context how their methods are QSAR methods. For example, it seems that the continuous data-driven descriptors (CDDD) is a QSAR method. Is that right? Maybe state it in the methods, just to tie in this introduction paragraph into the rest of the document. It seemed isolated and unclear the described methods were QSAR methods.”*

The reviewers correctly note that our approach aligns with QSAR methods. Traditionally, in QSAR, chemical structures are mapped to physicochemical properties or biological activities through quantitative relationships that describe how specific structural features influence these outcomes. The QSAR model is then used to predict the property or activity of new molecules. Three elements are needed to construct the QSAR model: (1) a robust algorithm (e.g., machine learning models) that can learn how to quantifiably relate chemical structures to their activities, (2) an informative numerical representation of these molecules, and (3) a high-quality training dataset from which patterns can be obtained.

In our study, we represented the molecules in our training set by Morgan2 fingerprints, latent (CDDD), and transformer-based descriptors (RoBERTa). These descriptors can serve as inputs for a QSAR framework (e.g., conformal predictor), but are not QSAR methods by themselves as the information on the biological activities is not part of the molecular encoding. To clarify this, we have revised the Methods section to explicitly state that our approach is part of the QSAR field (pages 4 and 28).

13. Reviewers: *“Figure 3: The figure descriptions for panels (a), (b), (c), and (d) seem to be scrambled, while panels (e) and (f) are accurately described. It seems that (a) describes panel (d), (b) describes panel (a), (c) describes panel (b), and (d) describes panel (c). To avoid confusion, please correct the correspondence between panel description and label.”*

We thank the reviewers for noting this inconsistency. We corrected the panel descriptions and their respective labels. The revised figure descriptions now accurately match the appropriate panels.

14. Reviewers: *“Figure 4: Could the authors provide docking scores for molecules 1 and 2 shown in panel (b)? This would allow the reader to gain an appreciation for where these molecules fall on the distribution shown in panel (a).”*

We thank the reviewers for this suggestion. We added the docking scores for compounds **1** and **2** in the score distribution in panel (a). In addition to this suggestion, we also show the dual docking scores of compound **5** in Figure 5c.

15. Reviewers: *“For Figure 4 A. There appear to be four curves (Blue, light blue, pink, red) not just one in blue as is stated in the figure caption. It is also hard to see the differences between these four plots as they are on top of one another, particularly for the left side of the curve. We assume that the colors correspond to those discussed in Figure 3B where colors represent different significance thresholds.”*

We thank the reviewers for noticing this inconsistency in our figure. The four curves represented different segments of the molecules prioritized by the conformal predictor, *i.e.*, top one million, followed by the next one million, and so on. The updated figure now represents an aggregated score distribution of the top five million molecules. The figure text accompanying this figure now accurately describes this information.

16. Reviewers: *“Text in some of the figures is too small to read, e.g. Figure 5 panels B and C. We would suggest making the font bigger in the legend for these two panels.”*

As requested by the reviewers, we have increased the font size in the legends for these panels to improve the readability.

17. Reviewers: *“We see in the supplemental document, Supplemental Table 1, the authors provide which residues are polarized (tarted), but it is not clear how they redistributed the partial charge. Could the author describe the polarization of residues (tarting)? Specifically, for which atoms are the partial charges altered?”*

We thank the reviewers for this suggestion and have further clarified the redistribution of the partial charges in Supplementary Figure S16.

Reviewer #3

The third reviewer appreciated our work and described our manuscript as *“interesting approach to add to the rapidly evolving field of ML-accelerated ultralarge docking concepts.”* Reviewer opinion of the manuscript is *“The manuscript is overall very well written and basically ready for publication, although I have some minor remarks that could improve it even further”*.

1. Reviewer: *“Since the authors state in several places that their CP framework runs with modest resources, I feel that a bit more space should be devoted to discussing where the AMCP ranks in terms of the computational expense in comparison to other recent approaches, e.g., the different deep learning approaches in ref. 10-12, but also other recently revisited, more classical, ML approaches making similar claims (e.g., DOI 10.1021/acs.jcim.3c01661). What architecture was used to perform this study, how much time and resources were spent for training and prediction steps? How do you anticipate your approach to scale with even larger libraries?”*

As suggested by the reviewer, we have compared our conformal predictor framework to other recently developed approaches, such as the deep learning methods (refs 13-15, previously refs 10-12), as well as other classical ML approaches (e.g., ref. 42).

(1) Training of conformal predictors. We evaluate several key parameters that directly affect the model training in our manuscript. As the conformal predictor aggregates outputs from adjacent machine learning classifiers, multiple independent models have to be trained. Our analysis shows that aggregation of five independently trained classifiers leads an optimal balance between the model performance and the final number of models to train. We also evaluated how the size of the training set affects the machine learning performance and found that models trained on one million molecules reach equivalent or better predictive power as methods that rely on active learning on datasets containing 11 million molecules (e.g., ref 14). We assessed the impact of alternative descriptors on the performance of the conformal predictor and found that Morgan2 fingerprints lead to similar or better results than models trained on latent or transformer-based descriptors (e.g., as in refs. 32 and 50). Finally, our benchmarks indicate that arguably simpler algorithms, such as CatBoost, can reach superior performances in comparison to more resource-intensive deep neural networks or large language models (e.g., as in refs. 13 and 15).

Integrating these algorithms, descriptors and hyperparameters, it takes on average 1410 seconds to train a conformal predictor, despite the need to train five classification models (Figure RL2, question 7 from reviewers 1-2). With respect to hardware, we had access to 2x Intel Xeon Gold 6130 CPUs @ 2.10 GHz. Although all our models can be trained on CPUs, training and predicting using the RoBERTa architecture strongly benefits from GPU usage. Accordingly, we allocated twelve cores of a single Nvidia Tesla T4 GPU with 1844 GiB memory for training. In this case, average training times amounted to 376685 seconds, which corresponds to approximately 21 GPU-hours per RoBERTa model. Considering the equivalent predictive power between CatBoost and RoBERTa models, CatBoost was selected because the training time was >250-fold faster.

(2) Prediction of ultra-large chemical libraries. Conformal predictors composed of CatBoost classifiers do not require GPU hardware for the prediction of large libraries (in comparison to ref. 13). We found that the framework's prediction time scales linearly in inference with the size of the library, making it well-suited for handling the continuous expansion of commercial chemical libraries. In practical terms, scoring of one billion compounds on a standard multi-core desktop computer can be completed under five hours, as shown in Table 2 in the revised manuscript. Using our framework, we were able to screen over 3.5 billion compounds with ease (115 CPU-hours) and allowed us to explore new strategies to identify novel ligands with tailored properties in one of the largest libraries ever prospectively evaluated by machine learning models.

(3) Data management. In addition to comparing the computational costs associated with training of and inference by machine learning models, we compare resource requirements between different molecular representations. We would argue that, from a practical point of view, bit-vectors (e.g., Morgan2 fingerprints) are the only type of descriptor that will be able to handle trillions of molecules long-term due to the ease of compression when represented as sparse vectors. Methods using RDKit descriptors and other numerical representations (e.g., latent descriptors or molecular graphs) necessitate on-the-fly featurization (e.g. ref 13), increasing the overall computational costs.

We have included a comparison to other approaches in the discussion section (pages 21-23) and a detailed description of the hardware used on page 30.

2. Reviewer: *“Since some readers might be interested in this as a way to speed up VS without being familiar with CPs, the authors might want to dedicate a half-sentence to explaining what a Mondrian CP is (i.e., mention that this CP is for working with class imbalance on the first mention).”*

We thank the reviewer for this suggestion and have added a brief explanation to the manuscript (pages 4 and 6) to clarify what a Mondrian CP is, with a focus on how they handle class imbalance.

3. Reviewer: *“Page 11: Fix reference 28 in this paragraph for consistency.”*

We thank the reviewer for noticing the inconsistency and have adjusted this reference accordingly.

4. Reviewer: *“It is of no major consequence, but the authors should clarify if they evaluate 3-5% of the full dataset of 235 million, or if they refer to 3-5% of the 234 million that were not used for training.”*

We thank the reviewer for this suggestion. As we exclude the training data from our objective test set, the final library for predictions contains 234 million molecules. We have clarified this in our manuscript (page 15).

5. Reviewer: *“Fig. 3b: The numbers in the significance dial inset are partly very hard to read with black font on top of dark red and blue. Can you change the font color to white in at least the two darkest slices?”*

We thank the reviewer for the feedback and have changed the font color to white in the two darkest slices as suggested by the reviewer (now Figure 3c).

6. Reviewer: *“Figs. 4b and 5ef: Since you are showing docking results as opposed to experimental structures, displaying hydrogens would make this more 'intuitive' to note for a reader. The figure captions should also tell the reader what they are looking at - something along the lines of: cartoon representation of D2R with key interacting residues in sticks representation, hydrogen bonding indicated by dashed orange line.”*

We have added polar hydrogens to the figure and added additional information to the legend (Figure 4b and Figure 5e-f).

7. Reviewer: *“I would recommend changing the deprecated sklearn in your setup.py to scikit-learn. As is, your code will otherwise cause an error during installation on relatively recent OS (and this has a really simple fix).”*

As suggested by the reviewer, we have updated the setup.py on our GitHub repository.

8. Reviewer: *“However, I would recommend that you add a simple example run to your README that really just lists the necessary commands to do preparation, run training and prediction. While your current README explains the key options and tweaks for each step, I found myself getting a bit lost when trying to find the commands to run a quick example, just to see whether the code works or not, so the most basic test run in the beginning of the README would really elevate things for me.”*

As recommended by the reviewer, we have added a simple example run to the beginning of the README file on our GitHub repository. The example section of the README file now lists the minimum necessary commands for preparation, training, and prediction.

Reviewer #4

The fourth reviewer appreciated our work and mentioned *“They conduct relevant benchmarking of their method assessing the role of molecular fingerprints, gradient boosted decision tree and deep-learning based ML method, and key method parameters including the model ensemble size, number of docked molecules, score-threshold, and sensitivity to noise.”* and considered our manuscript for publication *“Overall The manuscript text and figures are clear, and the conclusions and claims are overall well supported.”*

Major comments:

1. Reviewer: *“First, while they do cite prior work towards the problem of accelerating physics based virtual screening with ML, they there are at least 9 additional works that are highly comparable that they do not cite. I would like to better understand in what ways their method is superior to these others, e.g. in terms of performance, computational cost, robustness etc.”*

We thank the reviewer for this question, and there are indeed a large number of methods that focus on this important problem. We have included additional references suggested by the reviewer and present advantages of our method/work compared to other approaches below:

(1) Performance and robustness. The conformal predictor achieves similar or better performance than other available methods using less compute resources as we do not rely on active learning (e.g., as in ref. 13) and use gradient boosted trees instead of graph neural networks (e.g., as in ref. 14). Furthermore, we note that the conformal prediction framework is based on a mathematical proof and is, to the best of our knowledge, the only machine learning method other than Venn-Abers that can guarantee the error rates of the predictions to correspond to user-defined significance values. We benchmark our method on several types of descriptors and a larger number of targets than previous works (e.g., 13, 14, and 42). In our experience, the hyperparameters provided in this manuscript generalize well across the different protein targets and no resource-intensive parameter optimization is required for initial predictions. Based on the papers mentioned by the reviewer and those already cited, our prospective machine learning accelerated virtual screen of 3.5 billion compounds is one of the largest libraries ever evaluated.

(2) Computational cost. The conformal predictor models do not rely on GPU hardware for training (e.g, as in ref. 14) and can score one billion compounds on a standard multi-core desktop computer in less than 5 hours (see reviewers 1-2, question 8). In contrast to previous work that requires iterative training on 11 million compounds using active learning (e.g., ref. 15), we only use a single-iteration and achieve similar performance. CatBoost models are generally faster in training and predictions compared to the deep neural networks used in other methods (e.g., as in refs. 13 and 15).

(3) Scalability. In this study, we did a thorough comparison between different molecular descriptors and found that machine learning models trained on Morgan2 fingerprints have similar or better performances as those trained on latent or transformer-based descriptors (e.g., as in refs. 32 and 50). From a practical point of view, bit-vectors are the only feasible

type of descriptor that can store trillions of molecules long-term due to the ease of compression.

(4) Open science and accessibility. To enable the research field to perform screens of multi-billion-scale libraries and further development of methods in this area, we share code on our GitHub (<https://github.com/carlssonlab/conformalpredictor>).

Moreover, we provide access to several large data sets for benchmarking of methods. Several research groups (refs 13, 15, and 42) benchmark their methods using the same dataset generated by Lyu *et al.* 2019 (AmpC 96 million and D₄R 138 million compounds, ref. 5). While we recognize the need for established benchmarking datasets in this area of research, we found that the machine learning performance was strongly dependent on the protein target. If the field focuses on a small number of targets and the same datasets, the developed methods may not generalize well across protein targets. We expand the repertoire of drug targets by publicly sharing docking scores of 235 million molecules against eight unrelated protein targets (<https://doi.org/10.5281/zenodo.7953917>).

(5) Prospective ligand discovery. Our study remains among the few in which a new method was both developed and prospectively tested in experimental assays against therapeutic targets of interest. We are among the first to explore new strategies to tackle the challenge of identifying novel compounds with complex pharmacological profiles, such as multi-target ligands, and validate these predictions with experiments.

We agree that the number of available methods in this field of research is growing fast and have included additional references to other recent studies (ref 42). Because of Nature Computational Science's constraints on the number of references, we were not able to include all the references suggested by the reviewer. Some of the highlighted publications aim to traverse relatively small (million-scale sized) virtual libraries or use GPUs to accelerate docking without any use of machine learning and were therefore excluded.

2. Reviewer: *"Second, their use of conformal prediction is good, however they do not adequately explain why prior application of conformal prediction to learn chemoinformatic tasks does not naturally extend to their use case. Specifically from my reading of (Svensson et al., 2017) they also used very similar molecule encodings (RDKit descriptors vs. Morgan2) and ML based methods (Random Forest vs. CatBoost), but claim "Strategies to improve the virtual screening efficiency using the CP framework have been explored, but these workflows were not suitable for multi-billion-scale libraries and focused on traditional classifiers.", why is this? Additionally, I would appreciate it if the authors could more clearly articulate the advantages of the conformal prediction as many readers may not be familiar with it."*

Our work was indeed inspired by previous implementations of conformal prediction in cheminformatics, such as the work by Svensson *et al.* (ref. 20). However, while this workflow is suitable for prioritizing compounds from chemical libraries containing thousands of compounds for experimental evaluation, the approach is not suitable for handling multi-billion-scale databases.

Directly applying the approach suggested by Svensson *et al.* to multi-billion-scale libraries would require docking of the entire database, an obstacle that our work specifically aims to avoid. The core idea of Svensson's approach is that the top 1% of the library is experimentally evaluated, which remains impossible for make-on-demand databases as synthesis and testing of more than ten million compounds would be required. Testing a smaller number of compounds is an option, but training machine learning models on such extremely imbalanced datasets are unlikely to perform well on libraries containing billions of compounds. In fact, Svensson *et al.* actually show that their approach does not yield the error rate expected from the specified significance level due to the limited training data. Further increasing library sizes can be expected to have an even more negative impact on the validity. Consequently, the approach developed by Svensson *et al.* is most suitable for improving the efficiency of screens of libraries containing thousands of compounds, *e.g.*, those used in experimental high-throughput screening.

Finally, one could of course imagine modifying Svensson's approach to predict docking scores, an idea briefly mentioned by the authors. Notably, based on the results presented by Svensson *et al.*, their approach would require docking of approximately 10% of the database (>100 million compounds for a multi-billion-scale library). Such calculations are unfeasible for most research groups. Our work clearly shows that extending the conformal prediction approach to multi-billion-scale libraries is more challenging than simply increasing the size of the input database in previously established workflows. To enable the application of the conformal prediction to larger chemical libraries, we implemented several new techniques:

(1) Database reduction using the quality of information: Direct application of Svensson's approach would require docking of approximately 10% of the database (>100 million compounds for a multi-billion-scale library). In our manuscript, we explore several different strategies to solve this problem. Although conformal predictors with guaranteed validity require evaluation of molecules classified as {both} and {ones}, we find that lowering the significance level (Figure 3c) or sorting the predictions according to the quality of information metric prioritizes molecules with even better docking scores. As shown in Supplementary Figure S10, we additionally found that there is a correlation between docking rank and the quality of information, which consequently is a very useful metric in selection of smaller sets of compounds.

(2) Scalable generation of molecule encodings: It should also be noted that a limitation of Svensson's study is that topological fingerprints (*e.g.*, Morgan2) cannot be used for the selected benchmarking set (DUD-E), which is also noted by the authors in their paper. More importantly, there are key differences between RDKit descriptors and Morgan2 fingerprints that affect applications to multi-billion-scale libraries. Morgan2 fingerprints have the following advantages: (1) Arithmetic with bits leads to faster prediction times for large-scale libraries, and (2) bit-vectors can be densely compressed using a sparse representation enabling efficient storage. Morgan2 fingerprints are hence more suitable for applications involving databases containing billions of molecules. Due to the storage requirements and the continuous expansion of chemical libraries, RDKit descriptors and other numerical representations (*e.g.*, latent descriptors) would require on-the-fly featurization, increasing the overall computational costs.

(3) Public sharing of code: In contrast to Svensson *et al.* and several of the other groups working on conformal prediction in cheminformatics, we publicly share well-documented code on our GitHub repository. This is a crucial step for enabling the research field to perform screens of multi-billion-scale libraries and further development of methods.

We have improved our description of the advantages of our conformal prediction approach on pages 4, 6, 21, and 22.

3. Reviewer: *“Third, the authors should address why the predictive accuracy is less than what was found e.g. for D4R using a highly physics based virtual screening method in (Lyu et al., 2021). Specifically, does the proxy ML model hurt down performance? One concern is that ML based proxy methods may be focusing on the common chemotypes and ignoring rare chemotypes, effectively decreasing the diversity of predicted ligands, thus this would increase the similarity of selected hits and thus increase the variance in discovery. This can be directly measured e.g. through quantifying the diversity of the predicted vs. docking hits. Note this lack of diversity is partially apparent in the UMAP panel 3E the blue “Priority” points are much more concentrated than the red “Active train set” points.”*

These questions are interesting, but we are not entirely sure what work the reviewer is referring to as we do not cite any “Lyu et al., 2021” paper. In our response, we have assumed that the questions are regarding the results obtained for the D₄ dopamine receptor in ref. 5 (Lyu et al., 2019) and ref. 13 (Yang et al., 2021). The first study performs physics-based virtual screening screens of 138 million compounds to identify D₄ dopamine receptor ligands and the second papers uses machine learning to analyze the docking data from Lyu et al. We respond to the reviewer’s questions below:

(1) Is the “*predictive accuracy... less than what was found*” by Yang et al. (2021)?

The study from Yang et al. (ref. 13) reports two sets of molecular docking calculations against the D₄ dopamine receptor using a library of 138 million compounds. This data can be compared to the performance we obtained for another subtype, the D₂ dopamine receptor, in a docking screen of 235 million compounds. As both our study and Yang et al. use DOCK3.7, we compared the machine learning results using this data set. Wang et al. also performs screens using the program Glide and we note that the predictive accuracy of machine learning was better in this case. However, the difference in accuracy reflects that the performance of docking programs is target-dependent rather than the machine learning workflow. We therefore focused on comparing our conformal predictor framework to Yang’s active learning workflow for the DOCK3.7 data set.

In the case of the D₂ dopamine receptor, we trained a conformal predictor on one million molecules, corresponding to 0.4% of the library (235 million compounds). Similarly, Yang et al. trained a regressor using between 0.1% and 0.5% of their library of 138 million molecules. After evaluating the top 5% and 10% of the chemical library, where the machine learning models had the highest confidence, Yang et al identified 87% and 95% of the 10000 top-ranked molecules using their active-learning protocol. In comparison, our conformal predictor identified 90% and 96% (Figure 3d of this manuscript). The “*predictive accuracy*” obtained by Yang’s and our machine learning

workflow is hence comparable. Our analysis show that similar recall values can be obtained from machine learning models that were trained on similar sized fractions of the chemical library. Notably, our single-iteration workflow based on CatBoost and Morgan2 fingerprints can achieve such accuracies using far less computational resources than the graph-convolutional neural networks used by Yang *et al.*

(2) Is the “*predictive accuracy... less than what was found*” by Lyu *et al.* (2019)?

The study from Lyu *et al.* (ref. 5) reports prospective molecular docking screens of 138 million compounds against the D₄ dopamine receptor. Predicted ligands were synthesized and tested experimentally, yielding hit rates of 23-26%. In our study, a set of 31 compounds were selected from machine learning accelerated docking screens of 3.5 billion compounds, and two of these were experimentally confirmed to be D₂ ligands (hit rate of 6%). The hit rates from Lyu *et al.* are hence four-fold higher. Interestingly, a previous screen against the D₃R subtype resulted in an even higher hit rate despite that a considerable smaller library of 4.1 million compounds was used (56%, ref 45 and <https://doi.org/10.1124/mol.113.088054>). Several factors could contribute to these differences: (1) The screens focused on different subtypes (D₂ versus D_{3/4} receptors); (2) Our library contained compounds of higher molecular weight (rule-of-four) than that used by Lyu *et al.*; and (3) the structural differences between the used D₂ and D_{3/4} receptor structures are unexpectedly large. Given the large number of differences between the studies, we cannot draw any certain conclusion regarding the observed hit rates. However, the reviewer’s observation is interesting because one of the hopes of using larger libraries is that more potent compounds and higher hit rates will be obtained. Screens targeting the same receptor and under the same compound selection criteria will be required to answer this interesting question and recent results indicate that this is true (refs. 5 and 10). Although this is out of the scope of this work, we have added a section regarding these outstanding questions in the discussion (page 23).

(3) Does the machine learning approach lead to focus on “*the common chemotypes and ignoring rare chemotypes, effectively decreasing the diversity of predicted ligands*”

To answer this question, we compared our screens of 235 million compounds against the D₂ dopamine receptor using either molecular docking only or the machine learning workflow. We evaluated whether these methods yield top-scoring molecules with different levels of chemical diversity and examined the influence of virtual library size on this outcome.

Starting with the full library of 235 million compounds, we first generated multiple random subsets, reducing each subset’s size by half in successive iterations. For both virtual screening methods, we compared the number of unique Bemis-Murcko scaffolds among the top 1% of ranked molecules in each subset. Our results show that the conformal predictor identifies top-scoring compounds represented by fewer scaffolds, and this trend intensifies as the virtual library size increases (See below, Figure RL4a). This is an expected outcome, as these greedy models often exploit known, high-scoring molecular patterns rather than exploring broader chemical space for diverse scaffolds (*i.e.*, rare chemotypes). To further explore the difference in

chemical diversity, we calculated pairwise Tanimoto coefficients for the top 1% of molecules identified by large-scale physics-based docking and our machine-learning approach. The Tanimoto coefficient distributions indicated that, irrespective of library size, molecular docking produced top-scoring molecules with slightly higher chemical diversity than machine learning, although the difference was not statistically significant (paired t-test p-value = 0.99, See below: Figure RL4b).

It should be noted that, in practice, there are straightforward techniques to obtain a small set of diverse compounds for experimental evaluation. For example, we cluster the molecules based on their Morgan2 topological fingerprints after the explicit docking of the machine learning-prioritized compounds before visual inspection. We incorporated these new results on page 16 and in Supplementary Figure S11 of the revised manuscript.

Figure RL4. (a) Number of unique Bemis-Murcko scaffolds in the top-ranked (1%) compounds prioritized by explicit docking or the conformal predictor. (b) Distributions of pairwise Tanimoto coefficients in the top-ranked (1%) compounds prioritized by explicit docking or the conformal predictor.

4. Reviewer: “Page 3: In referencing the size of accessible chemical space, consider citing BioSolveIT’s KnowledgeSpace, as they claim they are enumerating 290,000,000,000,000 drug-like compounds. (Bellmann, 2022, 10.1021/acs.jcim.2c00334) Calculating and Optimizing Physicochemical Property Distributions of Large Combinatorial Fragment Spaces”

We thank the reviewer for this suggestion and now cite the paper by Bellmann *et al.* (ref. 4, <https://doi.org/10.1021/acs.jcim.2c00334>).

5. Reviewer: *“Therefore, there is an urgent need for more efficient virtual screening approaches able to evaluate multi-billion-scale libraries.” => I think this needs justification, as it's not obvious that larger compound libraries are better. You could cite "Modeling the expansion of virtual screening libraries", (Lyu et al., NatChemBio, 2023, DOI: 10.1038/s41589-022-01234-w), which I think is relevant and you cite later in the manuscript for a different claim.”*

We agree that it is important to substantiate the claim that larger compound libraries benefit both the experimental hit rate and the quality of these hits. To address this, we have introduced a brief explanation and the cited the paper from Lyu *et al.* in this section (ref. 10). We also mention this interesting topic in a new discussion section (page 23).

6. Reviewer: *“Traditionally, QSAR models have been trained on experimental data, but there is an increasing interest to predict which compounds in make-on-demand libraries are likely to receive favorable scores from computationally expensive virtual screening methods.” => You should also cite these additional methods for accelerating large-scale virtual screens”*

We thank the reviewer for this suggestion and agree that the number of available methods in this research domain is increasing fast. For this, we have included an additional reference to a recently published study (ref 42) later in our manuscript page 21. Because of Nature Computational Science's constraints on the number of references, we were not able to include all the references suggested by the reviewer. We have excluded the highlighted publications which aim to traverse relatively small (million-scale sized) virtual libraries or use GPUs to accelerate docking without any use of machine learning.

7. Reviewer: *“For citing conformal prediction, cite the original method and not just tutorials about the method.”*

We agree with the reviewer and have replaced citation to the original method (ref. 16).

UPPSALA
UNIVERSITET

SciLifeLab

January 20th, 2025

Dear Dr. McCardle,

We look forward to publishing our paper ("*Rapid Traversal of Ultralarge Chemical Space using Machine Learning Guided Docking Screens* ", Manuscript: NATCOMPUTSCI-24-1139A) in Nature Computational Science. We have addressed the additional comment regarding code availability. Please find below point-by-point responses to the referee with their comments in *Blue*.

Reviewer #2

The reviewer had additional suggestions on further improving the code availability: *"The authors mention the following in their methods section: "The scikit-learn 0.24.2 package was used to perform a stratified split the datasets in proper training sets (80% of training set), calibration sets (20% of training set), and test sets. The ratio between virtual actives and inactives was maintained in all sets. This procedure was repeated using different random seeds to obtain independent sets.""*

and *"Within the README file example, the authors mention that a separate file called "test_smiles.txt" should be specified to the program when making predictions, implying that the actual held-out "test set" splitting is not being done by the program itself, but by the user when they desire to make predictions on some test data. This aspect should be clarified to the user, so that they are aware that the stratified split is only being performed on the training data [which is being further split into "training (80%)" and "validation (20%)" (calibration) sets during K-fold cross-validation], and not the dataset as a whole (which would also include the held-out test data on which predictions are being performed)."*

Remarks on code availability

We thank the reviewer for this comment and have implemented the suggested changes in a new version of the repository's README file.